# A next-generation anti-CTLA-4 probody mitigates toxicity and enhances anti-tumor immunity in mice

Weian Cao[1,2,4], Junfan Chen[1,2,4], Yutong Fu[1,2,4], Haitao Jiang [1,2], Yu Gao[1,2], Huiming Huang[1,2], Yang-Xin Fu [1,2,3] ✉ & Wenyan Wang [1,2] ✉

CTLA-4 is a promising target for immune checkpoint inhibition in cancer therapy, with CTLA-4 blockade achieving prolonged overall survival for responding patients. However, the progressively elevated doses of anti-CTLA-4 agents, aimed at achieving better efficacy, result in increased toxicities, limiting their clinical applications. Here, we generate a prodrug design of the anti-CTLA-4 antibody, named ProCTLA-4, by folding the Fab fragment of the antibody in a tumor-associated protease-based manner. In preclinical mouse models, ProCTLA-4 effectively depletes suppressive regulatory T cells within the tumor microenvironment and enhances tumor-associated antigen-specific CD8+ T cell responses, while exhibiting reduced toxicity compared to currently available CTLA-4 blockade approaches. Furthermore, compared to the currently used Probody therapeutics for anti-CTLA-4 (BMS986288), ProCTLA-4 has more advantages in efficacy amplification, such as in poor immunogenic melanoma. Our design establishes an alternative paradigm for antibody agents that limits the emergence of immune-related adverse events (irAE) while increasing therapeutic efficacy.

Cytotoxic T lymphocyte antigen 4 (CTLA-4) is one of the important immune checkpoints on activated T cells. It shares the ligands of CD80/86 with CD28, which forms a bidirectional signal for regulating T cells. As the co-stimulatory receptor, CD28 strengthens the first signal that activates T cells, enabling T cells to further proliferate and differentiate. CTLA-4 can limit the activation signal of CD28 by competing for its ligands, CD80/CD86, with much higher affinity. Disrupting the binding of CD80/CD86 to the inhibitory receptor CTLA-4 to enhance their engagement with CD28 is critical for eliciting T cell responses in the tumor microenvironment (TME). Thus, the classic immune checkpoint blockade theory holds that the blockade of CTLA-4 releases CD80/CD86 from inhibitory sequestration to potentiate T cell-driven anti-tumor responses[1,2].

Furthermore, CTLA-4 is highly expressed on regulatory T cells (Treg) but relatively low on activated T cells within TME. Some studies have shown that the preferential depletion of Tregs within TME by CTLA-4 antibody is also an important anti-tumor mechanism, which depends on antibody-dependent cell-mediated cytotoxicity (ADCC) via Fc-receptor[3]. By eradicating Tregs, the immunosuppressive environment is converted, and the TME is driven towards a more pronounced anti-tumor activation state.

Tumor immunotherapy, and particularly strategies targeting immune checkpoints, has made remarkable progress in the clinic in recent years. Antibody therapies against CTLA-4, programmed cell death 1 (PD-1), and its ligand PD-L1 serve as the leading regimens[4]. While anti-PD-1 antibodies have been widely used in dozens of tumor types, ipilimumab (anti-CTLA-4, Ipi) is only approved for a few types of cancer, since its severe immune-related adverse events (irAE) reach 60% in patients with 3-4 grades at therapeutic doses[5]. Most observed toxicities are rash, hepatotoxicity, cytokine storm, colitis, or

[1]School of Basic Medical Sciences, Tsinghua University, Beijing, China. [2]State Key Laboratory of Molecular Oncology, School of Basic Medical Sciences, Tsinghua University, Beijing, China. [3]Changping Laboratory, Changping District, Beijing, China. [4]These authors contributed equally: Weian Cao, Junfan Chen, Yutong Fu. ✉e-mail: yangxinfu@tsinghua.edu.cn; wywang2022@tsinghua.edu.cn

cardiovascular alteration[5–7]. Reducing the dose could attenuate its toxicity but also diminish its efficacy. Conversely, increasing the dose makes it difficult for most patients to recover from severe irAEs. This has significantly limited the clinical application of the CTLA-4 antibody[8,9]. Once patients respond to CTLA-4 antibody and recover from the toxicity, almost 90% of them could have prolonged survival over ten years[10]. CTLA-4 antibody can synergize with PD-1 antibody to improve clinical efficacy, but concurrently induce more severe toxicity. Therefore, how to restrict the irAEs induced by CTLA-4 antibody while improving the therapeutic effect is a crucial issue that needs to be solved urgently.

A recent approach to overcoming irAEs of antibodies involves designing proteolytically mediated prodrugs, called Probody therapeutics. A probody consists of a 10–20 amino acid-long masking peptide linked to the N-terminal of the light chain of the antibody and a protease-cleavable linker as the connecting element. In peripheral tissues, the function of the antibody is blocked by the masking peptide from the target binding. Once the antibody reaches the TME, the linker will be cleaved by the proteases in the tumor and thereby release the antibody to the targets[11]. The CTLA-4 targeted probody with ADCC-enhanced Fc (BMS-986288) demonstrated reduced toxicity compared to Ipi[12,13]. The phase II clinical trial was completed in 2024 (NCT03994601). However, for reasons not explicitly stated, BMS was not intended to continue the development of BMS-986288 beyond the phase II study[14].

In this study, we introduce ProCTLA-4, an alternative prodrug design for the CTLA-4 antibody based on limiting the binding affinity via a protease-cleavable linker. ProCTLA-4 exhibits not only reduced toxicity but also enhanced anti-tumor efficacy in both murine and humanized models. This strategic approach for probody design has the potential to reduce the toxicity and enhance the efficacy of antibody-based therapeutics.

## Results

### The ProCTLA-4 antibody exhibits reduced toxicity
Concerning the limited blocking effort of peptide or induced additional antigen in vivo, we designed an advanced version of ProCTLA-4 with reduced affinity in the periphery, which can be restored in the TME. To limit the binding site targeted by the variable domain of heavy chain (VH) and light chain (VL), we attempted to fold the VH to link with the C-terminal of the constant domain of light chain (CL), which led to a distance between VH and VL. At the junction between CL and VH, we have inserted a matrix metalloproteinase (MMP) cutting site flanked with additional linkers in combination so that VH can be released by MMP protease cleavage in the TME (Fig. 1a and Supplementary Fig. 1a–c)[15]. Before examining the antigen-binding capacity of ProCTLA-4 antibody, we first tested the cleavage efficacy of the MMP linker. The ProCTLA-4 antibody was co-incubated with MMP14 protease for 24 h in vitro and then subjected to SDS-PAGE electrophoresis. As shown in Supplementary Fig. 1b, over 80% of the ProCTLA-4 antibody was cleaved by MMP14 protease. To investigate the binding properties, the binding of Ipi and ProCTLA-4 to a human CTLA-4 overexpressing cell line (hCTLA-4) was measured in the context of cleavage by MMP14. It was found that the affinity of ProCTLA-4 was roughly 14 times weaker compared to that of Ipi. However, the affinity of ProCTLA-4 post digestion by MMP14 was comparable to that of Ipi (Fig. 1b). This suggests that ProCTLA-4 inherently has a lower binding affinity, yet this affinity can be restored following MMP digestion. To test the in vivo drug distribution, we fluorescently labeled the ProCTLA-4 and monitored the fluorescence intensity at the tumor site following drug injection. In comparison with Ipi, the ProCTLA-4 antibody manifested a more concentrated and prolonged presence in the tumor site for up to 96 h after injection (Supplementary Fig. 1e, f), suggesting an enhanced local enrichment of ProCTLA-4 rather than its dissipation in the periphery. To evaluate the pharmacokinetics of

ProCTLA-4 and Ipi, antibodies were administered intraperitoneally to human CTLA4 knock-in (hCTLA4-KI) mice, and the concentration was monitored in serum within 12 days. As shown in Supplementary Fig. 1g, ProCTLA-4 reached the peak concentration quickly in 3 h and remained at the peak concentration for a much longer time than Ipi did in the following 6 days.

To compare their toxicity at the therapeutic dose (10 mg/Kg), we established a peripheral blood mononuclear cell (PBMC) transferred NSS mouse model to evaluate the toxicity of the antibodies. After continuous antibody administration, the Ipi antibody induced severe toxicity, which was manifested as weight loss exceeding 20%, cytokine storm with elevated IFNγ release in the serum, and a 100% mortality rate within 40 days after treatment. In contrast, the ProCTLA-4 antibody demonstrated remarkable safety, with no weight loss or cytokine storm, and a 100% survival rate (Fig. 1c–e). To further evaluate the potential autoimmune tissue damage, we compared ProCTLA-4 and Ipilimumab in young hCTLA4-KI mice combined with anti-PD-1 treatment to sensitize the mice to irAE as reported before[16]. With the anti-PD-1 combination, Ipi induced severe body weight loss and a high level of TNF in serum, but ProCTLA-4 did not (Fig. 1f, g). Furthermore, Ipi elicited severe multi-tissue inflammation, which was manifested as pancreatitis, colitis, pulmonary fibrosis, and hemorrhage. In contrast, ProCTLA-4 exhibited lower pathological scores representing improved tissue inflammation (Fig. 1h, i). Accordingly, the low-affinity ProCTLA-4 enhanced local enrichment of antibody in the tumor and showed reduced systemic toxicity.

### Low-affinity ProCTLA-4 exhibits enhanced anti-tumor efficacy
The reduced affinity led to a substantially decreased toxicity of ProCTLA-4 antibody, prompting us to question whether the anti-tumor efficacy had been compromised. To assess the efficacy, hCTLA4-KI mice were inoculated with immunogenic colon cancer (MC38) or poor immunogenic melanoma (B16F10) and then treated with Ipi or ProCTLA-4 antibody. In the MC38 model, tumors relapsed relatively early in the Ipi treatment group (Fig. 2a). In contrast, ProCTLA-4 therapy could result in a higher rate of tumor regression without relapse (Fig. 2a, b). Further, we tested the dose gradient of ProCTLA-4 in the MC38 tumor model. The EC50 was calculated by the proportion of regression upon ProCTLA-4 treatment in vivo in about 40 µg (Supplementary Fig. 2). While B16F10 is always considered as ICB therapy resistant[17], ProCTLA-4 still demonstrated greater effectiveness in tumor control compared to the Ipi treatment (Fig. 2c). To further explore the clinical application potential of this antibody, we established humanized mice model through CD34+ stem cell transplantation and subsequently inoculated A549 tumor cells. The Ipi antibody exhibited limited efficacy in tumor suppression, whereas the ProCTLA-4 antibody was capable of effectively controlling tumor growth (Fig. 2d). To elucidate the in vivo mechanism underlying ProCTLA-4, we additionally expressed the anti-mouse CTLA-4 antibody and its pro-form based on the sequence of clone 4F10[18] (Supplementary Fig. 1b). Both in the MC38 and CT26 tumor models, the pro-mCTLA-4 antibody exhibited superior tumor control efficacy compared to that of the mCTLA-4 antibody (Fig. 2e, f). These data demonstrated that the low-affinity ProCTLA-4 antibody possessed not only the advantage of improved safety but also enhanced anti-tumor efficacy.

### The ProCTLA-4 depletes ICOS-high, super-inhibitory Treg cells
Although the anti-CTLA-4 antibody exerts an immune checkpoint blockade effect, the selective depletion of Treg cells within TME rather than in the periphery might be crucial in balancing efficacy and toxicity of anti-CTLA-4. CTLA-4 is highly expressed on Treg cells, particularly on those within the TME[19]. Given the concentrated distribution of ProCTLA-4 in the tumor site, we examined the proportion of Treg cells in both TME and the peripheral tissues following ProCTLA-4 treatment.

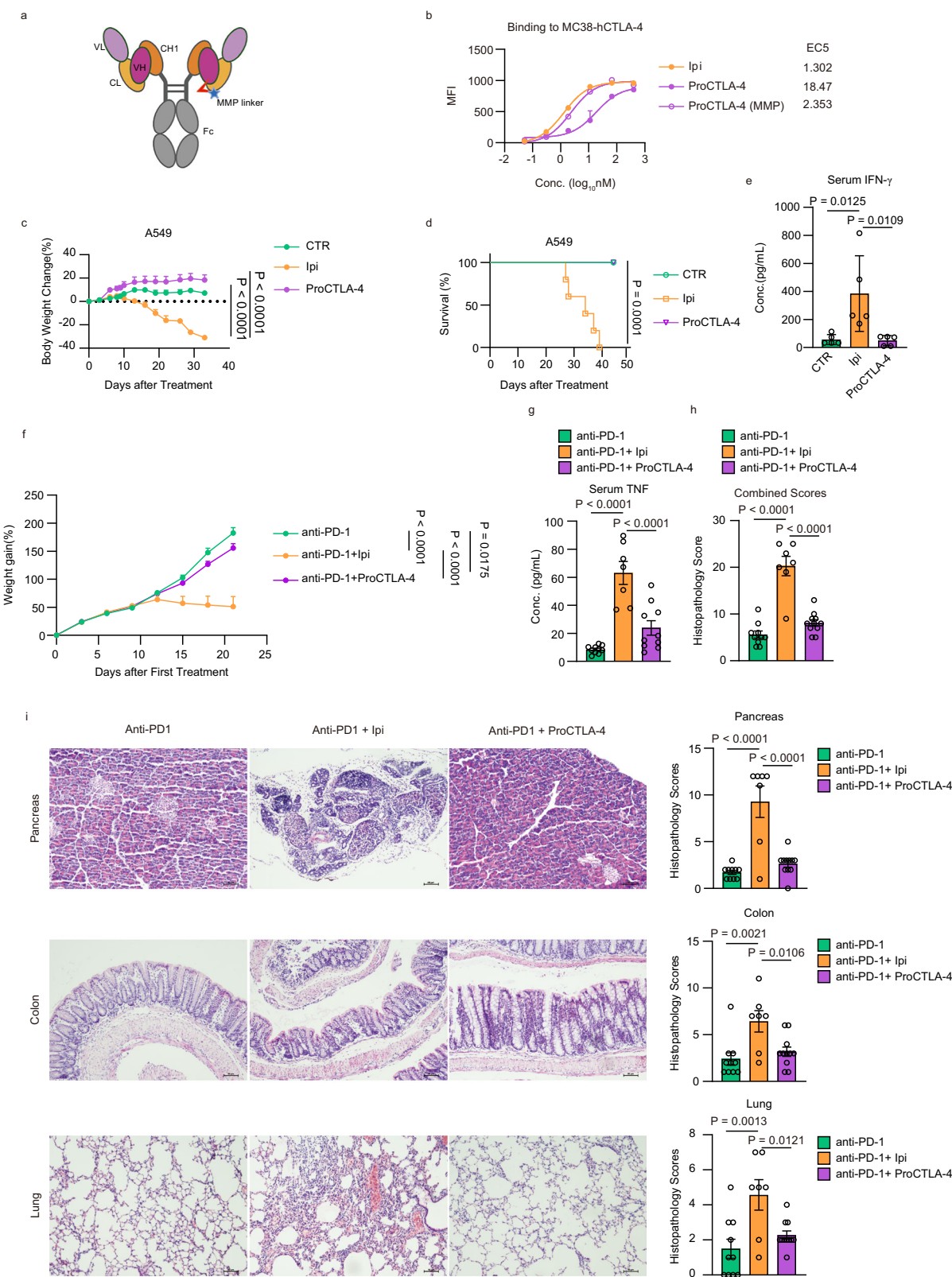

Compared with the CTLA-4 antibody, ProCTLA-4 led to a more enhanced depletion of Treg cells within the TME. However, the percentages of Treg cells in the draining lymph nodes (dLN) and spleen remained unaffected by either the anti-CTLA-4 or ProCTLA-4 antibody (Fig. 3a, b and Supplementary Fig. 3a). To dissect the alterations in the immune microenvironment induced by ProCTLA-4, we carried out single-cell sequencing of CD45+ cells in the TME. Following data pre-

processing and quality control, the cells were partitioned into six major clusters and classified by representative marker genes (Fig. 3c). Overall, ProCTLA-4 treatment induced more T/NK cells, neutrophils, and monocytes infiltration, compared to the control tumor, even more than anti-CTLA-4 treatment. In contrast, the frequency of tumor-associated macrophages showed a notable decrease in the ProCTLA-4-treated tumor (Fig. 3d).

**Fig. 1 | ProCTLA-4 antibody shows improved toxicity. a** Schematic structure of ProCTLA4 antibody, which was created with BioRender (https://BioRender.com/90qugj2). **b** Human CTLA-4-overexpressing MC38 cells (MC38-hCTLA-4) were incubated with serial dilutions of Ipilimumab (Ipi), ProCTLA-4 antibody and MMP-digested ProCTLA-4 antibody (ProCTLA-4(MMP)). Protein binding to MC38-hCTLA-4 was detected by flow cytometry. **c**–**e** A549 tumor-bearing male NSS mice were transferred 5 million human peripheral blood mononuclear cells (PBMC) via intravenous injection (i.v.) on day 8 after tumor inoculation and were intraperitoneally (i.p.) treated with PBS or 200 μg Ipi or equimolar ProCTLA4 (n = 5/group) on days 9, 12 and 15 after tumor inoculation. The percentage of body weight change (**c**), the percentage of survival curve (**d**), and the level of IFN-γ in serum (**e**) were measured. **f**–**i** 10-day-old female and male hCTLA-4 KI mice were treated i.p. with

anti-PD-1, anti-PD-1 plus Ipi, or anti-PD-1 plus ProCTLA-4 (n = 10/group), respectively, at a dose of 100 μg on days 10, 13, 16, and 19 after birth. Mice were euthanized on day 21 after the first treatment for analysis. The body weight change after treatment was measured (**f**). TNF (**g**) in serum was analyzed by CBA. Composite histopathology scores of the multi-tissues (**h**) are shown. Representative images of H&E-stained paraffin sections from the pancreas, colon and lung (**i**) are shown (anti-PD-1 (n = 10), anti-PD-1 plus Ipi (n = 7), or anti-PD-1 plus ProCTLA-4 (n = 10)). Scale bar, 50 μm. All data are representative of two or three independent experiments. Statistical analysis was performed using non-linear best fits (**b**), two-way ANOVA with Tukey's multiple comparisons test (**c**, **f**), log-rank (Mantel-Cox) test (**d**) or ordinary one-way ANOVA with multiple comparisons test (**e**, **g**, **h**).

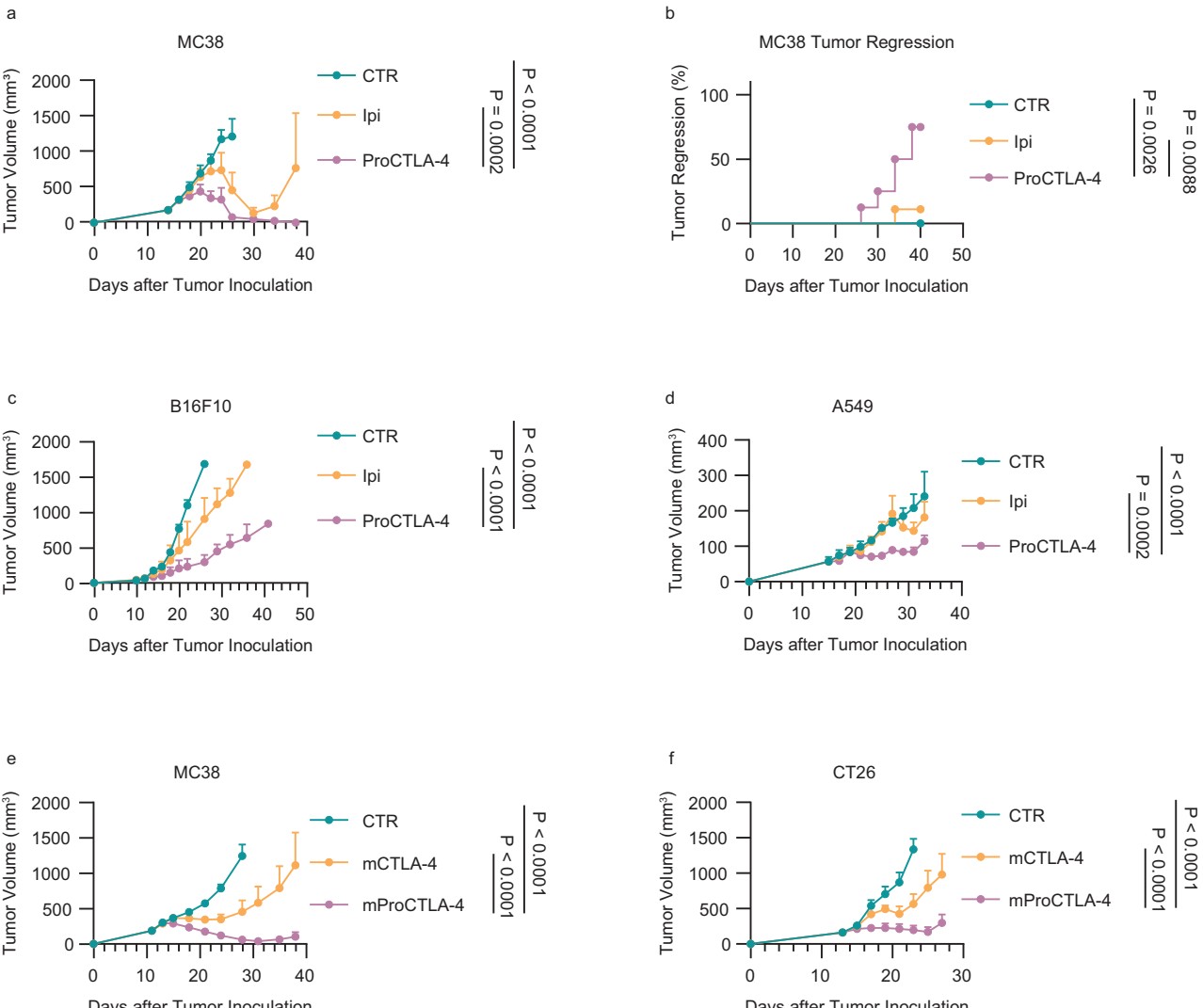

**Fig. 2 | ProCTLA-4 shows better efficacy in tumor controlling. a, b** MC38 tumor-bearing male hCTLA-4 KI mice were i.p. treated with PBS (CTR) (n = 8), 100 μg Ipilimumab (Ipi) (n = 9) or equimolar ProCTLA-4 (n = 8) on days 14, 17, and 20. The tumor growth (**a**) and the percentage of tumor regression (**b**) were recorded. **c** B16F10 tumor-bearing male hCTLA-4 KI mice were treated with CTR, 100 μg Ipi, or equimolar ProCTLA-4 (n = 5/group) on days 12, 15, 18, 21. The tumor growth was measured. **d** A549 tumor-bearing humanized male NSG-SGM3 mice engrafted with CD34[+] human hematopoietic cells were i.p. treated with CTR (n = 3), 100 μg Ipi (n = 4), or equimolar ProCTLA-4 (n = 4) on days 15, 18, 21 and 24. The tumor growth was monitored. **e** MC38 tumor-bearing male C57BL/6 mice were i.p. treated with

CTR, 100 μg anti-mouse CTLA4 antibody (mCTLA-4), or equimolar mouse ProCTLA-4 (mProCTLA-4) (n = 6/group) on days 11 and 200 μg mCTLA-4 or equimolar mProCTLA-4 on day14. The tumor growth was recorded. **f** CT26 tumor-bearing male BALB/c mice were treated with CTR, 100 μg mCTLA4, or equimolar mProCTLA-4 (n = 5/group) on day 13. Tumor growth was measured. Data in panels (**a**, **b**) are pulled from two independent experiments. Data in panels (**c**–**f**) are representative of two independent experiments. Statistical analysis was performed using two-way ANOVA with Tukey's multiple comparisons test (**a**, **c**–**f**), or log-rank (Mantel-Cox) test (**b**).

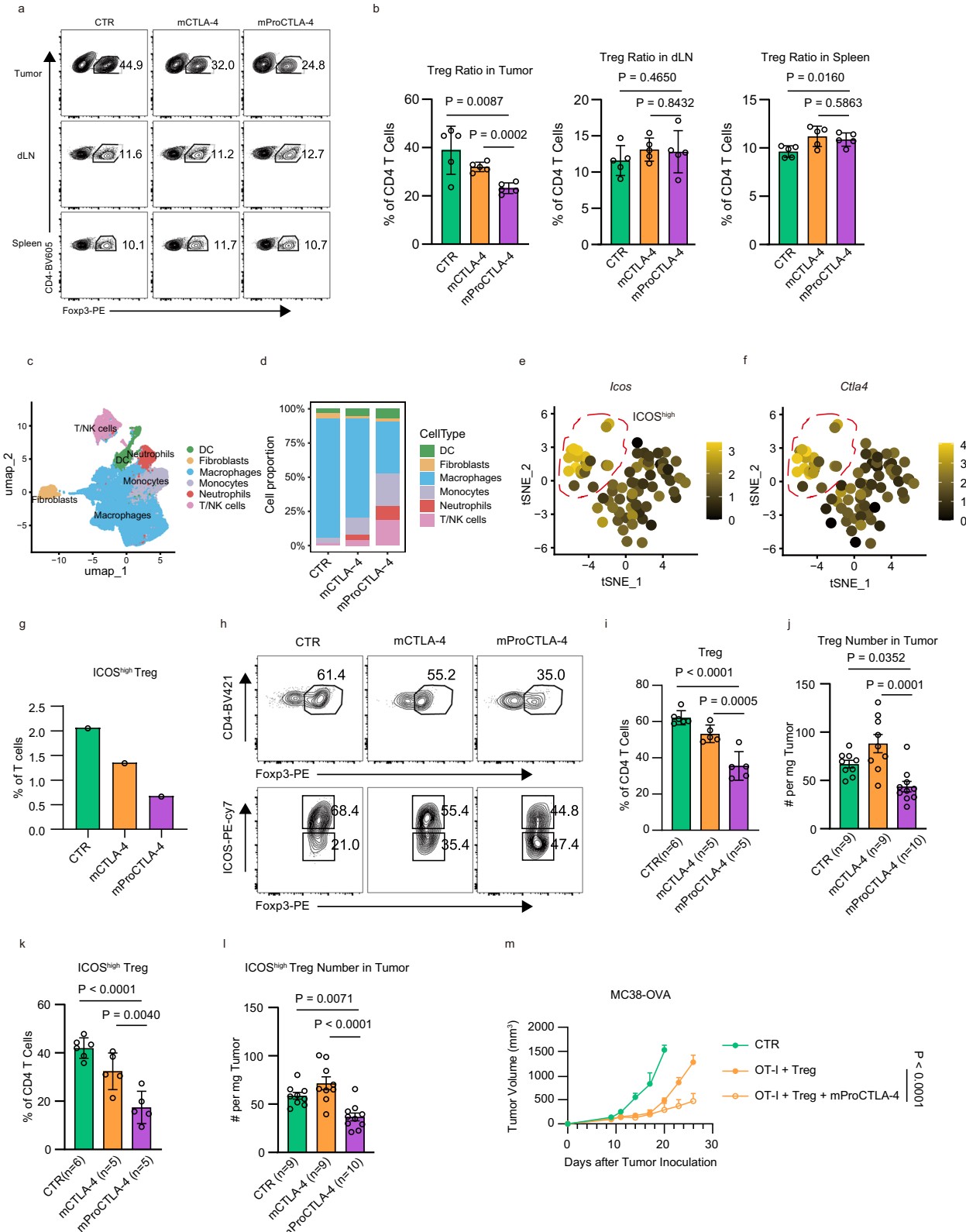

Treg cells were classified into two subsets according to their expression profiles, namely ICOS-high and ICOS-low expressing Treg cells (Fig. 3e). ICOS-high Treg cells are regarded as the super-inhibitory Tregs, which displayed higher proliferation and production of IL-10[20]. Since the ICOS-high Tregs also showed higher expression of CTLA-4 (Fig. 3f), the proportion of this subset was decreased upon ProCTLA-4 treatment compared to CTLA-4 therapy (Fig. 3g). To validate the

alterations in Treg cells, we examined the Treg cells within the TME following the antibody treatment. Our findings revealed that a low dose of ProCTLA-4 was still capable of depleting Treg cells, but the CTLA-4 antibody failed to do so (Fig. 3h–j). Notably, ProCTLA-4 exhibited a preference for depleting ICOS-high Treg cells both in proportion and absolute number (Fig. 3k, l), but did not impact the conventional CD4$^+$ T (CD4$_{conv}$) cell population (Supplementary

**Fig. 3 | ProCTLA-4 prefers to deplete super-inhibitory Treg in TME. a, b** MC38 tumor-bearing male mice were treated i.p. with CTR, 40 μg anti-mouse CTLA4 antibody (mCTLA-4), or equimolar mouse ProCTLA-4 (mProCTLA-4) (*n* = 5/group) on day 11 after tumor inoculation. 2 days later, Treg cells from the tumor, spleen, and dLN were analyzed by flow cytometry. The representative plot (**a**) and the frequency of Treg cells (**b**) from different treatment groups were shown. **c–g** Tumor-infiltrating CD45[+] immune cells isolated from MC38 tumor-bearing male mice with different treatment (CTR, mCTLA-4, mProCTLA-4) were analyzed by scRNA-seq. **c** UMAP visualization of merged total CD45[+] cells from three treatment groups. **d** Bar plot depicting the cell percentages of each cell type as a proportion of total cells. Feature plot illustrating gene expression of ICOS (**e**) and CTLA-4 (**f**) in Foxp3[+] Treg cells. **g** Bar plot depicting the cell percentages of ICOS[high] Treg cells within total CD3[+] T cells. **h–j** MC38 tumor-bearing male mice were treated i.p. with CTR, 10 μg mCTLA4, or equimolar mProCTLA4 on day 11 after tumor inoculation. 2 days after the treatment, the total Treg cells and ICOS[high] Treg cells (**h**) from tumors were analyzed by flow cytometry. The frequency and absolute number of total Treg cells (**i, j**) and ICOS[high] Treg cells (**k, l**) was quantified. **m** Female Rag1[-/-] mice (*n* = 5/group) were i.v. transferred with Tregs. 3 days post-transfer, the mice were challenged s.c. with MC38-OVA. Activated OT-I T cells were transferred 4 days later. 5 days after OT-I transfer, mice were treated with 100 μg mProCTLA-4 every three days for three times, and the tumor curve was monitored. Data in panels (**b, i–m**) representative of two independent experiments. Statistical analysis was performed using ordinary one-way ANOVA with multiple comparisons test (**b, i–l**), or two-way ANOVA with Tukey's multiple comparisons test (**m**).

Fig. 3b), as suggested by the single-cell RNA sequencing. To determine whether the depletion of ICOS-high Tregs was dependent, we constructed mProCTLA-4 with a mutant Fc that lacked the ADCC effect (mProCTLA-4$_{no\ ADCC}$)[21]. ProCTLA-4$_{no\ ADCC}$ failed to control MC38 tumor growth and showed no effects on ICOS-high Treg depletion (Supplementary Fig. 3c, d). To further confirm the Treg dependency in ProCTLA-4 treatment, OT-I cells with Tregs were transferred to Rag1[-/-] mice upon ProCTLA-4 treatment. Without the CD4$_{conv}$ T cells, ProCTLA-4 could still show sufficient tumor-controlling effect, which indicated that the efficacy of ProCTLA-4 depended on Treg cells (Fig. 3m). These data suggest that ProCTLA-4 might be more effective in reducing suppressive CTLA-4[+] Treg cells than regular anti-CTLA-4 antibody.

## ADCC-enhanced ProCTLA-4 displays augmented anti-tumor efficacy

Although ProCTLA-4 has demonstrated more potent anti-tumor activity by Tregs depletion, some tumors would still relapse in the long term after discontinuing treatment with the antibody. Therefore, it is necessary to improve depletion efficacy further. Fut8 is the crucial enzyme responsible for attaching fucose to an N-glycan. The existence of core fucose on the Fc region of antibodies has been demonstrated to impact ADCC effect and diminish the depletion efficacy of antibodies[22]. Hence, we employed Fut8 knockout CHO cells for the expression of antibodies, thereby obtaining the Ipi and ProCTLA-4 antibodies with enhanced ADCC activity. To verify the ADCC effect, the antibodies were co-incubated with ADCC reporter cells, and the downstream signals were monitored. Through evaluation based on EC50, it was revealed that the ProCTLA-4$_{ADCC}$ antibody exhibited a five-fold augmentation in ADCC effect in comparison to ProCTLA-4, even a 200-fold enhancement over the Ipi and Ipi$_{ADCC}$ antibodies (Fig. 4a). Subsequently, we investigated the anti-tumor efficacy of the antibodies with enhanced ADCC activity in the MC38 and B16F10 tumor models. In MC38 tumors, low dosage of ProCTLA-4$_{ADCC}$ manifested more potent tumor-controlling capabilities than either the formal ProCTLA-4 or the Ipi$_{ADCC}$, enabling the tumors to achieve complete regression (Fig. 4b). In the case of the "cold" tumor B16F10, ProCTLA-4$_{ADCC}$ managed to keep the tumor in a progression-free state for over 30 days, performing even better than the ADCC-enhanced Ipi antibody (Fig. 4c). Furthermore, in the humanized mice model, ProCTLA-4$_{ADCC}$ still showed the strongest tumor-controlling ability (Fig. 4d).

The improvement of antibody activity raises concerns about its toxicity. In these reconstituted NSS mice, both Ipi and Ipi$_{ADCC}$ led to an extreme loss of body weight and ultimately resulted in 100% mortality. In contrast, neither ProCTLA-4 nor ProCTLA-4$_{ADCC}$ triggered any weight loss, and the mice remained in total survival (Fig. 4e, f). Moreover, we also evaluated the toxicity in the PBMC-transferred NSS mouse model. The Ipi$_{ADCC}$ antibody manifested over 30% weight loss, a cytokine storm characterized by elevated IFNγ release, and 100% mortality. In contrast, ProCTLA-4$_{ADCC}$ antibody elicited no observable weight loss, no occurrence of cytokine storm, and a 100% survival rate was maintained (Supplementary Fig. 4a–c). The peripheral tissues of

the kidney, liver, and lung were isolated to examine pathological changes. Upon treatment with Ipi$_{ADCC}$, more pronounced inflammatory infiltration was detected in the kidney and liver. In comparison, no such infiltration was seen in the mice treated with ProCTLA-4$_{ADCC}$. In the lung tissue of the Ipi$_{ADCC}$-treated mice, lung consolidation and hemorrhagic effusion were present, whereas in the lungs of those administered with ProCTLA-4$_{ADCC}$, the alveoli were much clearer and the level of inflammation was significantly lower (Supplementary Fig. 4d). Taking these into account, we further boosted the anti-tumor efficacy of the ProCTLA-4 antibody by augmenting its ADCC activity. More importantly, the ProCTLA-4$_{ADCC}$ antibody retained its therapeutic safety with no observable toxicity.

## The ProCTLA-4 elicits a stronger antigen-specific T cell response

To verify the T cell subset on which the effect of ProCTLA-4 depends, we conducted depletion of CD4 and CD8 positive cells during ProCTLA-4 treatment. Depletion of CD4[+] T cells failed to impair the efficacy of ProCTLA-4. However, the absence of CD8[+] T cells invalidated the anti-tumor effect of ProCTLA-4 (Fig. 5a). These results suggested that the function of ProCTLA-4 was dependent on CD8[+] T cell response. Subsequently, we analyzed the CD8[+] T cells using the single-cell RNA sequencing data. Based on the expression profiles of distinct CD8[+] T cell phenotypes, we conducted further subpopulation analysis and identified four exhausted T subsets (Tex), naïve, proliferating, and memory-like subsets (Fig. 5b, c)[23-25]. The IFN-responsive Tex and Teff-memory subsets were more abundant in ProCTLA-4-treated mice, both in terms of average number and proportion. In contrast, the intermediate Tex subset was decreased (Fig. 5d). By RNA velocity analysis, two differentiation branches were identified: the IFN-responsive branch and the Teff-memory branch. Interestingly, the intermediate Tex cells were located in positions with higher differentiation potential to both branches. Upon ProCTLA-4 treatment, CD8[+] T cells preferred to transmit to IFN-responsive Tex or Teff-memory cells (Fig. 5e). To verify the antigen-specific response of CD8[+] T cells, we carried out the enzyme-linked immunosorbent spot assay (ELISpot) to detect the antigen-specific IFNγ producing cells within the draining LN. Treatment with CTLA-4 antibody showed comparable IFNγ[+] spots to those of the control group, whereas ProCTLA-4 induced a greater number of IFNγ[+]CD8[+] T cells in both MC38 (Supplementary Fig. 5a, b) and B16-OVA (Fig. 5f) tumor models. To test the memory branch of differentiation, mice that had experienced tumor regression following antibody treatment were subsequently rechallenged with a higher tumor load. In the case of the Ipi treatment, over 70% of the rechallenged tumors initially grew in the early stage and then shrank, which did not achieve complete regression. By contrast, in the mice treated with ProCTLA-4, over 80% of the rechallenged tumors failed to grow at all (Fig. 5g). Accordingly, more antigen-specific memory CD8[+] T cells were detected in the draining LNs with ProCTLA-4 treatment compared to CTLA-4 (Fig. 5h, i and Supplementary Fig. 5c). This indicated a stronger memory response elicited by the ProCTLA-4 treatment than that of Ipi.

To test if increased memory-like T cells at TME could lead to more effective anti-tumor at distal site, we administered ProCTLA-4 at a very

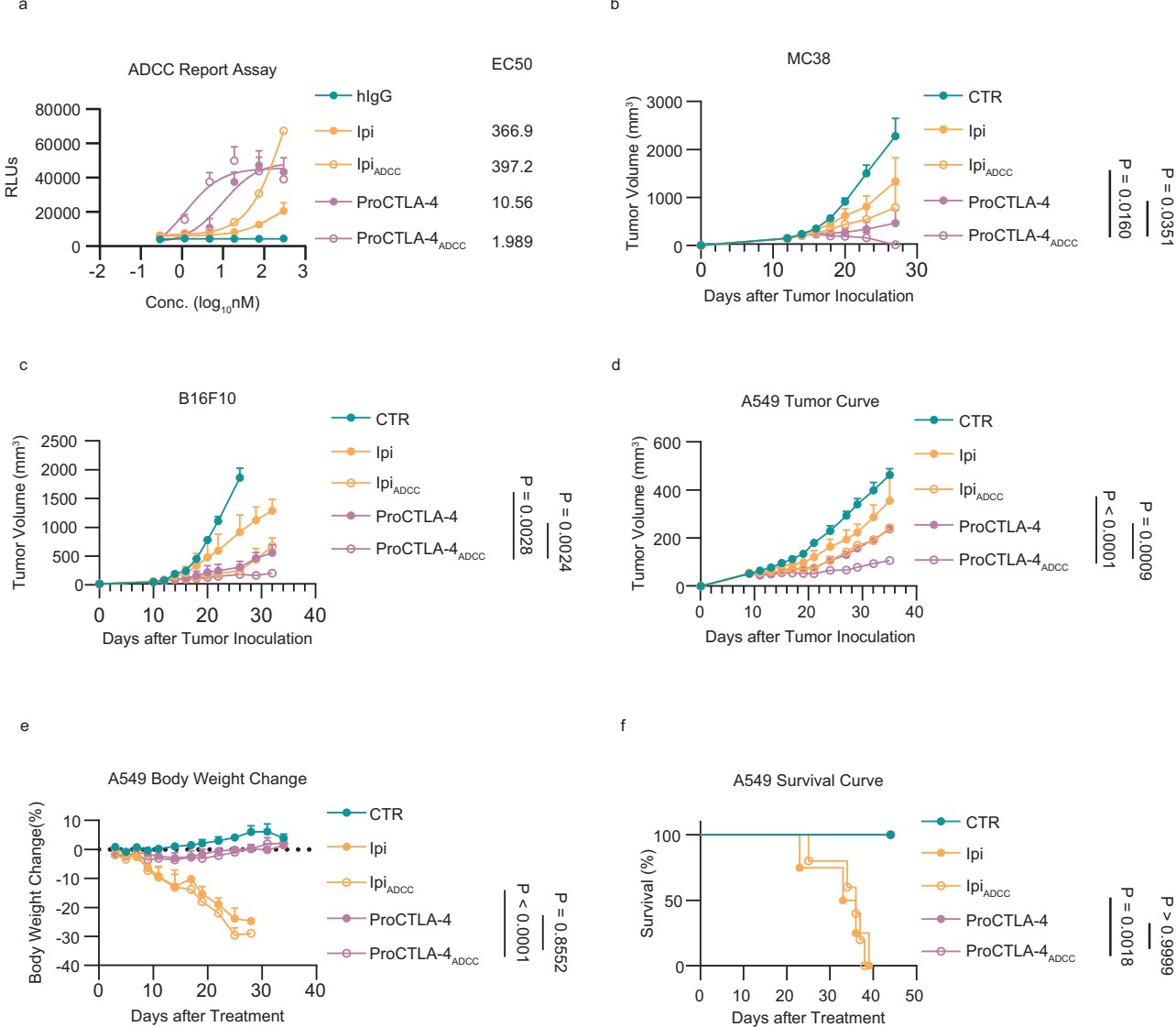

**Fig. 4 | ProCTLA-4 with ADCC enhanced Fc is boosted for anti-tumor efficacy.**
**a** MC38-hCTLA-4 cells were incubated with serial dilutions of Ipilimumab (Ipi), ADCC-enhanced Ipilimumab (Ipi$_{ADCC}$), ProCTLA-4, or ADCC-enhanced ProCTLA-4 (ProCTLA-4$_{ADCC}$) for 1 h. Jurkat-LuciaTM NFAT-CD16 effector cells were then co-incubated with MC38-hCTLA-4 cells for 6 h. ADCC response was assessed by Glo-max Multi Plus. **b** MC38 tumor-bearing male hCTLA-4 KI mice were i.p. treated with CTR ($n = 9$), 40 µg Ipi ($n = 10$), equimolar Ipi$_{ADCC}$ ($n = 9$), ProCTLA-4 ($n = 9$), or ProCTLA-4$_{ADCC}$ ($n = 9$) on days 12 and 15. The tumor growth was monitored. **c** B16F10 tumor-bearing male hCTLA-4 KI mice were i.p. treated with CTR ($n = 5$), 100 µg Ipi ($n = 5$), equimolar Ipi$_{ADCC}$ ($n = 4$), ProCTLA-4 ($n = 5$), or ProCTLA-4$_{ADCC}$

($n = 4$) on days 12, 15, 18 and 21, and the tumor growth was measured. **d–f** A549 tumor-bearing humanized male NSG-SGM3 mice engrafted with CD34$^+$ human hematopoietic cells were i.p. treated with CTR ($n = 5$), 100 µg Ipi ($n = 4$), equimolar Ipi$_{ADCC}$ ($n = 5$), ProCTLA-4 ($n = 4$), or ProCTLA-4$_{ADCC}$ ($n = 5$) on days 10, 13, and 16. The tumor growth (**d**), percentage of body weight change (**e**), and percentage of survival (**f**) were monitored. Data in panel (**b**) are shown as mean ± s.e.m. and are pulled from two independent experiments. Data in panels (**c–g**) are representative of two independent experiments. Statistical analysis was performed using non-linear best fits (**a**), two-way ANOVA with Tukey's multiple comparisons test (**b–e**) or log-rank (Mantel-Cox) test (**f**).

low dose intratumorally to the local tumor, in combination with FTY720 to block the lymphocytes trafficking from the LN to distant sites (Fig. 5j). The blockade of lymphocyte trafficking to tumor did not impair the efficacy of ProCTLA-4 against the local tumor, indicating that the existed lymphocytes within TME were sufficient to respond effectively to ProCTLA-4 treatment (Fig. 5k). In contrast, the distal tumor could not be controlled following FTY720 blockade, suggesting that the control of the distal tumor by local ProCTLA-4 administration was reliant on the trafficking of TILs through the draining LN to distal site (Fig. 5l). Taken together, our findings revealed that ProCTLA-4 antibody induced more memory and effector antigen-responsive CD8$^+$ T cells at TME that are essential for local and distal tumors as well as preventing relapse.

## The ProCTLA-4 showed more advantage over the Probody

Peptide-based Probody, such as BMS-986288, is the current state-of-the-art design. To compare these two CTLA-4 prodrugs, we first detected the binding affinity of ProCTLA-4 and BMS-986288 both before and after the protease digestion. The BMS-986288 showed a lower binding affinity compared to that of ProCTLA-4, indicating better blockade efficacy of BMS-986288. Following their specific protease digestion, the affinity of both ProCTLA-4 and BMS-986288 was comparable to that of Ipi. Notably, a lower concentration of urokinase-type plasminogen activator (uPA, 5:1) for pretreatment could not restore the affinity of BMS-986288, while the higher concentration of uPA (2:1) could do so (Fig. 6a). In both MC38 and B16F10 tumor models, ProCTLA-4 demonstrated a more potent anti-

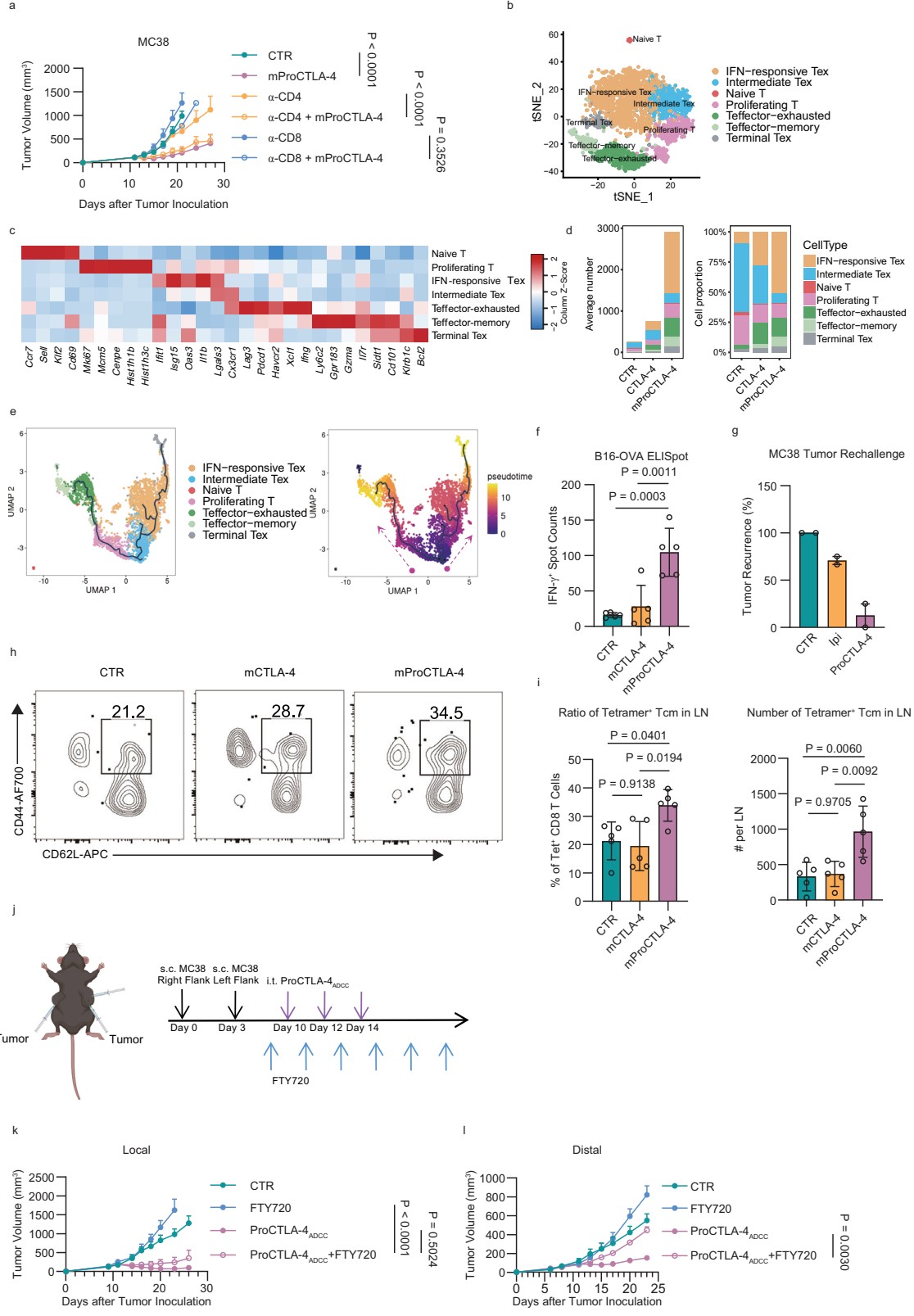

tumor capacity in comparison to BMS-986288 (Fig. 6b, c). Nevertheless, in the PBMCs transferred model, neither ProCTLA-4 nor BMS-986288 induced body weight loss or mortality, unlike what was observed in mice with Ipi treatment (Fig. 6d, e). These findings indicated that ProCTLA-4 and BMS-986288 had comparable reduced toxicity. Taken together, the ProCTLA-4 showed a similar safety profile to BMS-986288, yet exhibited a greater advantage in anti-

tumor efficacy, especially in overcoming ICB-resistant tumors, compared to BMS-986288.

## Discussion

CTLA-4 is one of the most highly promising targets in tumor immune therapy. Remarkably, patients who respond to anti-CTLA-4 often exhibit prolonged survival over ten years. However, with the wide

**Fig. 5 | ProCTLA-4 promotes Teffector and memory-like CD8⁺ T cell responses.** **a** MC38 tumor-bearing male C57BL/6 mice were i.p. treated with PBS (CTR), 40 μg mProCTLA-4 ($n = 5$) on days 12, 15, and 18. For CD4⁺ T or CD8⁺ T cell depletion, mice were i.p. injected with 200 μg α-CD4 or α-CD8 antibodies since day 11 post tumor inoculation in every 3 days for four injections in total. **b**–**e** scRNA-seq of CD8⁺ T cells isolated from total CD45⁺ cells for subset analysis. **b** tSNE visualization of CD8⁺ T cells colored by subsets. **c** Heatmap of average gene expression of marker genes in different CD8⁺ T cell subsets. **d** Bar plot depicting the cell number of each CD8 T cell subsets and their percentages as a proportion of total CD8 T cells. **e** CD8 T cells were ordered into a branched pseudotime trajectory using Monocle3. The UMAP embedding showed cells colored by subsets (left) or by their estimated pseudotime values (right). Black lines indicated the inferred branched trajectory. **f** B16-OVA tumor-bearing male mice were i.p. treated with PBS, 40 μg mCTLA-4, or equimolar mProCTLA-4 ($n = 5$/group) on days 10 and 13. Lymphocytes from the draining LN

were collected at the endpoint for the ELISpot assay. **g** MC38 tumor-bearing male hCTLA-4 KI mice were i.p. treated with CTR, 40 μg Ipi, or equimolar ProCTLA-4 ($n = 5$/group) on days 12 and 15. Cured mice were rechallenged with MC38 tumor cells on the opposite flank at 8 weeks post the initial inoculation. Rechallenged tumors reached 40 mm³ in volume was regarded as present tumors, and the percentage of tumor recurrence was calculated. **h**–**i**, Tetramer⁺ CD8⁺ central memory T cells (CD44⁺ CD62L⁺) in dLN were analyzed for frequency and absolute numbers ($n = 5$/group). **j**–**l**, Dual-flank MC38 tumor experiment was conducted (**j**), the tumor growth curve on the right flank (**k**) and the left flank (**l**) were monitored. Data in panels (**a**, **f**, **h**–**l**) are representative of two independent experiments. Data in panels **g** are pulled from two independent experiments. Statistical analysis was performed using two-way ANOVA with Tukey's multiple comparisons test (**a**, **k**, **l**), or ordinary one-way ANOVA with multiple comparisons test (**f**, **i**). The figure in panel (**j**) was created with BioRender (https://BioRender.com/90qugj2).

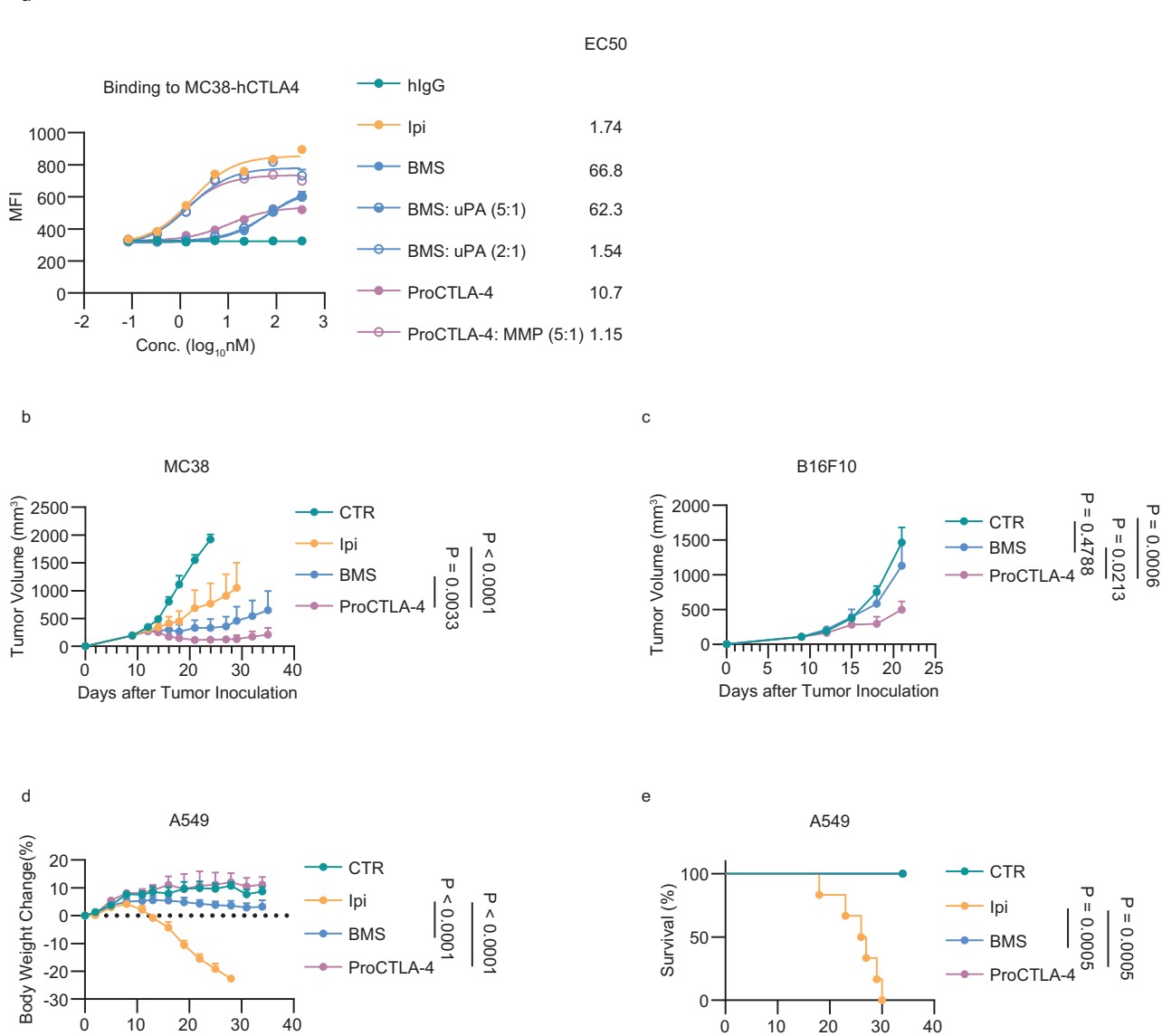

**Fig. 6 | The advantage of ProCTLA-4 compared to the Probody therapeutics.** **a** MC38-hCTLA-4 cells were incubated with serial dilutions of Ipilimumab (Ipi), ProCTLA-4, MMP-digested ProCTLA-4 at a 5:1 molar ratio ProCTLA-4/MMP (ProCTLA-4: MMP (5:1)), BMS, or uPA-digested BMS at 5:1 or 2:1 molar ratio BMS/UPA (BMS: uPA (5:1) or BMS: uPA (2:1)). Protein binding to MC38-hCTLA-4 was detected by flow cytometry. **b** MC38 tumor-bearing male hCTLA-4 KI mice were i.p. treated with PBS (CTR) ($n = 4$), 100 μg Ipi ($n = 4$), equimolar ProCTLA-4 ($n = 4$), or BMS ($n = 5$). The tumor growth was monitored. **c** B16F10 tumor-bearing male hCTLA-4 KI mice were i.p. treated with PBS (CTR), 150 μg BMS, or equimolar ProCTLA-4 ($n = 6$/

group) on days 9, 12, and 15. The tumor growth was monitored. **d**, **e** A549 tumor-bearing male NSS mice were transferred 5 million human peripheral blood mononuclear cells (PBMC) via intravenous injection (i.v.) on day 8 after tumor inoculation and were treated with CTR ($n = 5$), 200 μg Ipi ($n = 6$), equimolar ProCTLA-4 ($n = 6$), ProCTLA-4$_{ADCC}$ ($n = 6$), or equimolar BMS ($n = 6$) on days 10, 13, 16, 19, and 22. The percentage of body weight change (**d**) and the percentage of survival (**e**) was monitored. All data are representative of two independent experiments. Statistical analysis was performed using non-linear best fits (**a**), two-way ANOVA with Tukey's multiple comparisons test (**b**–**d**) or log-rank (Mantel-Cox) test (**e**).

usage and increased dosage, its severe toxicity always limits its application at therapeutic range, leading to few responders and the failure of most clinical trials. As a result of binding to the peripheral tissues, the anti-CTLA-4 antibody increases the T cell responses to induce irAEs. Meanwhile, excessive peripheral consumption reduces the concentration of antibody reaching the tumor site, thus also affecting the efficacy[26]. To avoid toxicity, physicians prefer to reduce the dosage of Ipilimumab, which, as a consequence, also impairs the therapeutic effect. Therefore, there is an urgent need to find a strategy to increase the dosage for better anti-tumor efficacy while limiting the peripheral toxicity of anti-CTLA-4 antibody.

There are several approaches to mitigating irAEs associated with the CTLA-4 antibody. A dual variable domain (DVD) immunoglobulin of anti-CTLA4 was designed to block the CTLA-4-binding domain by an outer tumor-targeting anti-prostate stem cell antigen (PSCA) domain with an MT-SP1 cleavage linker. This class of anti-CTLA4 antibody acted in a tumor-targeting way by enhancing antitumor activities while minimizing potential toxicities[27]. One study showed that engineered CTLA-4 antibody variants with increased pH sensitivity by preventing antibody-triggered lysosomal CTLA-4 downregulation dramatically attenuate irAE[16]. Another approach is the design of masking peptide-based probodies, a type of conditionally activated antibody prodrug that is supposedly cleaved intratumorally to release the whole antibody. In this study, the folded Fab of the antibody is an optimized design for the pro-form of Ipilimumab. We directly linked the VL-CL and VH-CH in a line sequence, which physically heightened the distance between VL and VH. To release the VH from CL linkage back to forming the normal antibody, we screened the various sequences for MMP14 proteases in the connection between CL and VH spaced with additional linkers. Based on this design, the ProCTLA-4 was highly blocked in binding to CTLA-4, which could be restored by MMP14 digestion. ProCTLA-4 limited cytokine storm and long-term autoimmune tissue damage and protected the mice from excessive weight loss or even death. On the contrary, the control CTLA-4 antibody exhibited severe toxicity, leading to higher mortality. Ipilimumab and BMS-986288 are the best-known anti-CTLA4 antibody and its related prodrug. ProCTLA-4 shows advantages in efficacy compared to Ipi and BMS-986288, not only in the immunogenic model, but also in the ICB-resistant tumor.

The ProCTLA-4 antibody preferred to bind to intratumoral Tregs, which express a much higher level of CTLA-4 than peripheral Tregs and other TILs within the tumor. Among Tregs, the ICOS[high] Tregs is considered as the "super inhibitory" Treg, which has stronger proliferative capacity and produces more inhibitory cytokine IL-10[20]. According to the single-cell RNA sequencing, we found that ICOS[high] Tregs and CTLA-4[high] Tregs were the same subset. ProCTLA-4 specifically depleted the ICOS[high]CTLA-4[high] Tregs within tumor depending on FcγR engagement, which resulted in converting the suppressive TME to T cells-accumulated TME. The ProCTLA-4[ADCC] with ADCC-enhanced Fc further strengthened its anti-tumor ability without increasing the risk of toxicity.

The T cell response is always essential for tumor control upon immune therapy. Among all the CD8[+] T cells within the tumor post-therapy, the IFN-responsive Tex was increased, but the intermediate Tex was decreased upon ProCTLA-4 treatment. The pseudotime speculation indicated that the intermediate Tex converted to IFN-responsive Tex and also to memory-like T effectors on the other path. The IFN-responsive Tex was characterized by IFN stimulator *Ifit1* and *Isg15*, which could induce type I IFN and stimulate the cross priming of dendritic cells (DC) and CD8[+] T cells. It was consistent with the DCs induced within TILs (Fig. 3d) and consequently the antigen-specific CD8[+] T cells accumulation by ProCTLA-4. Furthermore, more memory-like T effectors were observed in the ProCTLA-4-treated tumor and draining LN. This might provide an explanation for the better control of the rechallenged tumor by ProCTLA-4. Consistently, the distal

tumor could be effectively controlled, depending on memory-like lymphocytes trafficking through dLN. These results bring us two important inspirations. ProCTLA-4 not only depleted Tregs to alter the immune suppressive TME, but even more stimulated CD8[+] T cells towards more activated and responsive state, which consequently enhanced the anti-tumor efficacy. More essentially, ProCTLA-4 promoted and maintained more memory-like T effectors, which protected the host from tumor metastasis and relapse. In that case, ProCTLA-4 is expected not only to address the primary tumor but also to eliminate metastatic tumors to improve survival.

Taken together, we construct a next-generation of probody for the anti-CTLA-4 antibody, which displays reassuringly reduced toxicity and facilitates anti-tumor capacity, especially in ICB-resistant cases. This strategy for probody holds great potential to overcome the limitations posed by toxicity, which is associated with the application of all agonistic antibodies. This would open up a new prospect for immunotherapy and bring benefits to a large number of patients.

## Methods
### Mice
C57BL/6 mice (Stock Number: N000013) and Balb/c mice (Stock Number: 211) were purchased from GemPharmatech and Beijing Vital River Laboratory Animal Technology Co.Ltd. The hCTLA-4 KI mice (Stock Number: NM-HU-00014) were purchased from Shanghai Model Organisms Center, Inc. NSG-SGM3 mice (Stock Number: 013062) were purchased from the Jackson Laboratory. Rag1[-/-] mice were from Xiaohuan Guo's lab (Tsinghua University). Foxp3-YFP mice were from Yuncai Liu's lab (Tsinghua University). OT-I mice were from Xin Lin's lab (Tsinghua University). All mice were maintained under specific pathogen-free conditions with 10/14 dark/light cycle, 20–26 °C, and 30–70% humidity. Animal care and experiments were approved by the Animal Care and Use Committees (IACUC) of Tsinghua University. Experimental animals and control animals were bred separately. For euthanasia, the mice were euthanized by $CO_2$ according to the approved protocols. The maximum size of the tumor was controlled to less than 1500 mm³ (calculated by length*width*height/2) in volume according to the IACUC protocol.

### Cell lines and reagents
B16F10 (Cat#CRL-6475), CT26 (Cat#CRL-2638), and A549 (Cat#CCL-185) cell lines were purchased from the American Type Culture Collection. The MC38 cell line (Cat#305223) was purchased from Cytion. The MC38-hCTLA-4 cell line was constructed by transduction with lentivirus encoding the human CTLA-4. These cells were cultured in 5% $CO_2$ at 37 °C and maintained in Dulbecco's modified Eagle medium (DMEM) (Cat. Code: CM1003) supplemented with 10 % heat-inactivated FBS (Cat. Code: 10099-141 C), 2 mM L-glutamine, 100 μ/ml penicillin-streptomycin (Cat. Code: 15140163). The Freestyle 293 F cell line (R79007) was purchased from Invitrogen and was cultured in 8 % $CO_2$ at 37 °C in SMM 293-TII medium (Cat. Code: M293TII-1L). The Jurkat-Lucia NFAT CD16 cell line (Cat. Code: jktl-nfat-cd16) was purchased from InvivoGen and was cultured in IMDM (Cat. Code: 12440053) supplied with 10 % heat-inactivated FBS, 2 mM L-glutamine, 25 mM HEPES, and 100 μ/ml penicillin-streptomycin. Plasmids of all therapeutic antibodies were purchased from GENERAL BIOL. FTY720 (Cat. Code: SML0700) was purchased from Sigma-Aldrich. Anti-CD8 antibody (Lyt 3.2, BE0223), anti-CD4 antibody (GK1.5, BE0003-1) were purchased from BioXcell.

### Protein cleavage and binding affinity assay
MMP14 (R&D Systems) (Cat. Code: 918-MP-010) was activated according to the manufacturer's instructions. The human ProCTLA-4 antibody was cleaved by activated MMP14 in assay buffer containing 50 mM Tris-HCl (pH = 8.5), 3 mM $CaCl_2$, and 1 μM $ZnCl_2$. BMS-986288

was cleaved by uPA (Orileaf) in assay buffer containing 50 mM Tris-HCl (pH = 7.4), 150 mM NaCl, and 0.05 % Tween 20.

The MC38-hCTLA-4 cell line was first incubated with Ipilimumab, human ProCTLA-4 antibody (cleaved or not), BMS-986288 (cleaved or not), or human IgG with 5-fold serial dilution from 300 mM for 30 min at 4 °C, followed by incubated with PE anti-human IgG Fc antibody (1:200, 410708, Biolegend) for 30 min at 4 °C. The affinity was assessed by flow cytometry.

## Biodistribution

7-week-old hCTLA-4 KI mice were subcutaneously inoculated with $5 \times 10^5$ MC38 tumor cells. After 11 days, mice were randomly assigned into groups and intravenously injected with 40 μg Ipilimumab conjugated with cy5 or equimolar ProCTLA-4 conjugated with cy5. The accumulation of individual antibodies in tumors was detected by IVIS LUMINA III at 48 h, 72 h, and 96 h after injection.

## ADCC report assay

MC38-hCTLA-4 cells were incubated with serially diluted Ipilimumab, ADCC-enhanced Ipilimumab, ProCTLA-4 antibody (cleaved or not), ADCC-enhanced ProCTLA-4 antibody (cleaved or not), or human IgG at 37 °C in a $CO_2$ incubator for 1 h. Then, MC38-hCTLA-4 cells were incubated with Jurkat-Lucia NFAT CD16 cells at 37 °C in a $CO_2$ incubator for 6 h. After incubation, 20 μL supernatant was transferred into a 96-well white (opaque) plate and 50 μL of QUANTI-Luc 4 Reagent (Cat. Code: rep-qlc4lg1) working solution was added per well. NFAT activation, reflecting the induced ADCC response, was assessed by determining Lucia luciferase activity in the supernatant. The relative light unit (RLU) was measured by Glomax Multi Plus.

## Tumor growth and treatment

7-week-old mice were subcutaneously inoculated with $5 \times 10^5$ MC38, $2 \times 10^5$ B16F10, $5 \times 10^5$ MC38-OVA, or $3 \times 10^5$ CT26 tumor cells on the right flank. For the double-flank tumor model, $5 \times 10^5$ MC38 tumor cells were subcutaneously inoculated on the right flank of mice. After 3 days, $2.5 \times 10^5$ MC38 were s.c. inoculated on the left flank of mice. When tumors were established, mice were randomly grouped and were intraperitoneally or intratumorally treated with the indicated antibodies, respectively. For immune cell depletion, mice were intraperitoneally injected with 200 μg anti-CD4 antibody or 200 μg anti-CD8 antibody starting from one day prior to the first treatment and then once every three days for a total of four doses. For FTY720 blockade, mice were intraperitoneally injected with 20 μg FTY720 starting from one day before the first treatment and then once every other day for a total of five times. For the rechallenging study, $4 \times 10^6$ MC38 tumors were subcutaneously injected on the left flank of mice 8 weeks after the initial tumor challenge. The tumor volume was measured every two or three days and calculated as length*width*height/2. The humane endpoints are defined as a tumor volume ≤ 1500 mm³ for the MC38, ≤ 1000 mm³ for the B16F10, and ≤ 1000 mm³ for the CT26.

## Humanized mouse tumor model

For the PBMC-humanized mouse model, 7-week-old NSS mice were transferred 5 million human PBMCs (JUNX-BIO, CNHBC025C) via intravenous injection and were subcutaneously inoculated with $2 \times 10^6$ A549 cells on day 8 post PBMCs transferred. For the human CD34⁺ cells-humanized mouse tumor model, 4-week-old NSS mice were treated by 5.5 Gy × 2 irradiation and then transferred with $1 \times 10^5$ human CD34⁺ cells (JUNX-BIO, HSC010). After assessing the human cells proportion in peripheral blood by flow cytometry, successfully humanized mice were subcutaneously inoculated with $2 \times 10^6$ A549 cells. When tumors were established, mice were randomly assigned to groups and intraperitoneally injected with the indicated antibodies.

## Toxicity evaluation

For monitoring body weight change, the weight of the mice at the beginning of the treatment was set as the initial body weight. In the following days, the weight of the mice was measured every three days. The weight change was calculated as (measured body weight − initial body weight)/initial body weight * 100 %. For measuring cytokine levels in serum, the blood of mice was collected 24 h after the first treatment, and the serum was separated. The concentration of cytokines in serum was measured by BD Cytometric Bead Array (CBA) Mouse Inflammation Kit (Cat. Code: 552364) and Human Th1/Th2/Th17 CBA Kit (Cat. Code: 560484).

## Pharmacokinetics of antibodies

7-week-old hCTLA-4 KI mice were randomly assigned into two groups and intraperitoneally injected with 10 μg Ipilimumab or equimolar ProCTLA-4 antibody. The blood was collected at the indicated time point. The concentration of human IgG in serum was measured by enzyme-linked immunosorbent assay (ELISA).

## ELISA

Microtiter plates (Corning Costar) were coated with 5 μg/mL capture antibody (AffiniPure Goat Anti-Human IgG, Fcγ fragment specific) overnight at 4 °C. After washing and blocking, diluted serum from treated mice was added and incubated at 37 °C for 1.5 h. Then, the plate was washed, and horseradish peroxidase (HRP)-conjugated goat anti-human IgG (Fcγ fragment specific) was added and incubated at room temperature for 50 min. After the last wash, 100 μL TMB substrate was added, and the reaction was terminated by adding 50 μL 2 N $H_2SO_4$. The plate was read at 450 nm (minus 630 nm for wavelength correction) using a BioTek microplate reader.

## Mouse tissue processing

Tumors, dLNs, and spleens were dissected from euthanized tumor-bearing mice. For TIL isolation, tumor tissues were surgically minced in RPMI-1640 medium containing 3 % (v/v) FBS, digested by 1 mg/mL Collagenase IV (Sigma-Aldrich, C5138), 20 μ/mL DNase I (Sigma-Aldrich, D5205), and 0.1 mg/mL Hyaluronidase V (Sigma-Aldrich, H6254) for 1 h at 37 °C. After digestion, the tumor tissues were passed through 70 μm filters and centrifuged at $500 \times g$ for 5 min at room temperature. The cell pellets were used for the following experiments.

For mouse spleen lymphocyte isolation, the spleens were disrupted in PBS, passed through 70 μm filters, and centrifuged at $500 \times g$ for 3 min. The resulting cell pellets were re-suspended with 1 mL 1 × RBC lysis buffer (Invitrogen, 00-4333-57), incubated at room temperature for 1 min, and centrifuged at $500 \times g$ for 3 min after adding 14 mL PBS. The cell pellets were used for the following experiments.

For dLN lymphocyte isolation, the dLNs were disrupted in PBS, passed through 70 μm filters, and centrifuged at $500 \times g$ for 3 min. The cell pellets were used for the following experiments.

## Flow cytometry

Mouse primary cells isolated from tumors, dLNs, and spleen were first stained with eBioscience Fixable Viability Dye eFlour 506 (Invitrogen 65-0806-14) according to the manufacturer's instructions. Subsequent surface marker staining was performed in PBS containing 2 % (v/v) FBS for 45 min at 4 °C. For intracellular Foxp3 staining, the eBioscience Foxp3/Transcription Factor Staining Buffer Set was used according to the manufacturer's instructions. The following fluorescent conjugated-labeled antibodies were used: APC-eFluor 780-anti-CD45 (30-F11, 47-0451-82, 1:400, eBioscience), PE-anti-Foxp3 (FJK-16s, 12-5773-82, 1:400, eBioscience); Brilliant Violet 605-anti-CD4 (GK1.5, 100451, 1:400, BioLegend), Brilliant Violet 421-anti-CD4 (GK1.5, 100438, 1:400, BioLegend), FITC-anti-CD90.2 (Thy-1.2, 105306, 1:400, BioLegend), PE/Cyanine7-anti-CD278 (ICOS) (C398.4 A, 313520, 1:400, BioLegend), AF700-anti-CD44 (IM7, 103026, 1:400, BioLegend), APC-anti-CD62L

(MEL14, 104428, 1:400, BioLegend); T-Select H-2KbOVA Tetramer-SIINFEKL-PE (H-2kb, TS-5001-1C, 1:50, MBL), FITC -anti-mCD8 (KT15, K0227-4, 1:50, MBL). Samples were analyzed on a FACS Fortessa flow cytometer (BD Biosciences). Data were analyzed using FlowJo software.

### irAE model

10-day-old hCTLA-4 KI mice were treated with the indicated antibodies at a dose of 100 μg/mouse/injection every 3 days for a total of four doses. The body weight change of the mice was measured, and the organs from the mice were collected for H&E staining and histopathology analysis.

### H&E staining and histopathology analysis

Formalin-fixed paraffin-embedded tissue (5 μm) slides were deparaffinized with xylene, ethanol and deionized water. Slides were then stained with hematoxylin (Fisher Chemical, Cat. No. SH26500D) and excess signal was destained with acid ethanol. Eosin (Poly Scientific, Cat. No. s176) was applied to slides and washed with 95% ethanol. Stained slides were dehydrated with ethanol and xylene, and scanned using a Nikon upright microscopy imaging system. H&E sections were scored double blind. Lung samples were scored as follows: 0, normal; 1, minor perivascular inflammation; 2, increased perivascular and peribronchial inflammation; 3, severe perivascular, peribronchial and interstitial inflammation. Pancreas samples were scored as follows: 0, no inflammation; 1, mild perivascular inflammation; 2, moderate perivascular and intralobular inflammation; 3, severe inflammation; 4, severe inflammation and acinar necrosis. Colon samples were scored as follows: 0, no inflammation; 1, multifocal, mucosal infiltration; 2, diffuse, mucosal infiltration ± involvement of the submucosa; 3, moderate mucosal infiltration and submucosal inflammation; 4, severe mucosal and submucosal inflammation, hyperplasia of crypts ± loss of goblet cells, erosions. Individual or combined scores of all organs examined are shown.

### Preparation of CD45.1 OT-I, Foxp3-YFP Treg cells and adoptive T cell transfer

Spleens from CD45.1 OT-I mice were mechanically disrupted and ground through a 70 μm strainer (Biologix). Red blood cells (RBC) were lysed by 1 × RBC Lysis Buffer (eBioscience). CD45.1 OT-I were negatively selected from the splenocytes by MojoSort Mouse CD8 T Cell Isolation Kit (Biolegend) according to the manufacturer's instructions. Purified CD45.1 OT-I cells were resuspended at $1.5 \times 10^6$ per mL in RPMI1640 supplemented with 10 % FBS, β-mercaptoethanol (Gibco, 21985023), 100 μ/ml penicillin-streptomycin, non-essential amino acids (Gibco, 10370021), sodium pyruvate (Gibco, 11360070), HEPES (Gibco, 15630080), and glutaMax (Gibco, 35050061). T cells were plated in a 6-well plate with coated 5 μg/mL αCD3 (eBioscience, 14-0032-85) and 1 μg/mL αCD28 (eBioscience, 14-0281-85) supplied with 20 ng/mL recombinant mouse IL-2 (PeproTech, 212-12-20UG). After 72 h activation, 1 million OT-I cells were adoptively transferred to Rag1$^{-/-}$ mice per mouse.

For isolation of Foxp3-YFP Treg cells. Splenocytes from Foxp3-YFP mice were separated, and total CD4 T cells were purified by MojoSort Mouse CD4 T Cell Isolation Kit (Biolegend, 480033). Foxp3-YFP Treg cells were then sorted by FACS Aria SORP based on the fluorescence of GFP. 0.1 million Foxp3-YFP Treg cells were adoptively transferred to Rag1$^{-/-}$ mice per mouse.

### Single-cell RNA-seq library preparation

C57B6/J mice were inoculated 5*10$^5$ MC38 tumor cells, and tumor-bearing mice were treated with PBS (CTR group) or 100 μg ADCC-enhanced mouse CTLA-4 antibody (CTLA-4 group) and equimolar ADCC-enhanced ProCTLA-4 antibody (ProCTLA-4 group) on days 11 and 14 post inoculation ($n = 5$ for each group). MC38 tumors

were collected on day 17 post-inoculation for scRNA-seq library preparation. Each group homogenized 5 samples for single scRNA-seq processing.

Tumor tissues were dissected and enzymatically digested using the gentleMACS (Miltenyi) according to the manufacturer's instruction. Single-cell suspensions were stained with antibodies against 7AAD/CD45 for FACS sorting on a BD Melody instrument. Sorted live CD45$^+$ immune cells were loaded onto a GemCode Single-Cell Instrument (10 x Genomics, Pleasanton, CA) by using Single Cell 3' Library and Gel Bead Kit V4 (10x Genomics), and subsequent steps were performed according to manufacturer instructions. Single-cell RNA libraries were sequenced by an Illumina NovaSeq 6000 sequencer with 150 bp paired-end reads (performed by Analytical Bioscience Limited, Beijing). The scRNA-seq data generated were aligned and quantified using the Cell Ranger Single-Cell toolkit (v.8.0.0) against the mm10 mouse reference genome. Preliminary filtered data generated from Cell Ranger were used for downstream analysis.

### Single-cell RNA sequencing analysis

scRNA-seq data analysis was performed in R (v.4.3.3) using the Seurat (v.5.0.3) package. Three samples (CTR, CTLA-4, ProCTLA-4) were loaded and merged. Data were then normalized and identified variable features using the Seurat NormalizeData and FindVariableFeatures. Cells with fewer than 200 RNA features detected, greater than 50000 RNA counts detected, and greater than 5 % mitochondrial RNA content were excluded from analysis. We then scaled the data and ran PCA before initial clustering and dimension reduction. Cell types were predicted using SingleR (v.2.4.1) based on murine immune cell reference data (ImmGenData) from Celldex (v.1.12.0). The T/NK cluster was isolated and reanalyzed with the same process as above. CD8 T cell cluster was identified as the CD3e$^+$ CD8$^+$ Klrd1- population, and the Treg cell cluster was identified as CD3e$^+$CD4$^+$Foxp3$^+$. CD8 T cell subsets were annotated by the average expression level of marker genes, referring to literature[23–25]. Pseudotime trajectory prediction was analyzed by the Monocle3 (v.1.3.7) package.

The scRNA data were processed using Cell Ranger v.8.0.0 (https://www.10xgenomics.com) and were analysed with the R package Seurat v.5.0.3 (https://satijalab.org/seurat), R Studio v.2023.12.1 (https://posit.co/products/open-source/rstudio) and ggplot2 v.3.5.1 (https://github.com/tidyverse/ggplot2/releases).

### IFN-γ ELISpot

5 × 10$^5$ MC38 tumor cells or B16-OVA were subcutaneously injected on the right flank of C57BL/6 mice. Mice were intraperitoneally treated with PBS, 40 μg mouse CTLA-4 antibody, or equimolar mProCTLA-4 antibody on days 10 and 13. The MC38 tumor-bearing mice were euthanized on day 21, and the B16-OVA tumor-bearing mice were euthanized on day 18. Lymphocytes in the draining LN were collected for the ELISpot assay. Lymphocytes (5 × 10$^5$) were seeded in each well, supplied with the matched dead tumor cells subjected to repeated freeze-thaw cycles (5 × 10$^4$) or 5 μg/ml SIINFEKL peptide (OVA257−264) to stimulate the tumor-specific T cells. After 24 h, the ELISpot assay was performed using the IFN-γ ELISpot kit (BD Bioscience, 551083) according to the manufacturer's instructions. IFN-γ spots were enumerated with the Elispot Reader.

### Quantification and Statistical Analysis

The sample sizes (n), probability (p) value, and the specific statistical test for each experiment were indicated in the figure legends. Statistical significance was tested by two-sided one-way ANOVA with Tukey's test adjusted for multiple group comparisons or two-way ANOVA with multiple comparisons for curve comparisons using the GraphPad Prism 8.0 program. Data from such experiments are presented as

mean values ± SEM; $P < 0.05$ was considered significant. No data were excluded from statistical analysis. The data presented were representative of at least two independent experiments.

**Reporting summary**

Further information on research design is available in the Nature Portfolio Reporting Summary linked to this article.

## Data availability

All data are included in the Supplementary Information or available from the authors, as are unique reagents used in this Article. The raw numbers for charts and graphs are available in the Source Data file whenever possible. The single-cell RNA-seq data have been deposited on Gene Expression Omnibus with the accession number GSE305582. Source data are provided in this paper.

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

## Acknowledgements

We are grateful for the support provided by the State Key Laboratory of Molecular Oncology, the Animal Facility, Tsinghua University. We appreciate the support from the National Natural Science Foundation of China (82250710684 to YX.F., 32370967 to W.W.).

## Author contributions

W.W. and YX.F. conceived and designed the study and prepared the manuscript. W.C., J.C., and Y.F. performed all experiments and assisted in data analysis. H.J., Y.G., and H.H. participated in some experiments.

## Competing interests

The authors declare no competing interests.
