## [Transparent Peer Review file · Nature Communications]

A next-generation anti-CTLA-4 probody mitigates toxicity and enhances anti-tumor immunity in mice

Corresponding Author: Professor Yang-Xin Fu

Version 0:

Reviewer comments:

Reviewer #1

(Remarks to the Author)

In this manuscript, Weian Cao et al. present a novel probody design (ProCTLA-4) for anti-CTLA-4 therapy, aiming to enhance tumor-specific activation while reducing systemic toxicity. By linking the VH domain to the CL domain via an MMP-cleavable linker, the authors introduce an alternative strategy to traditional peptide-masked probody therapeutics (e.g., BMS-986288). The study demonstrates a reduced percentage of FoxP3+ cells within the CD4+ T cell population in the tumor microenvironment (TME), particularly targeting ICOS-high Tregs, and highlights the superior anti-tumor efficacy of ProCTLA-4 compared to Ipilimumab. Additionally, an ADCC-enhanced version (ProCTLA-4ADCC) is shown to further improve efficacy without increasing toxicity. While the antibody format is novel, certain aspects require further clarification and validation:

Major Comments:

1. The manuscript highlights the induction of memory-like CD8+ T cells, but it does not indicate when the mice were rechallenged in Figure 5g. This information is crucial for evaluating whether the experiment is assessing immune memory.
2. While ProCTLA-4 demonstrates slightly higher antitumor activity in an in vivo MC38 experiment, its advantage over BMS-986288 remains unclear since both share the same mechanism of action. It is unlikely that this novel format can overcome the limitations that led to the discontinuation of BMS-986288.
3. The assumption that ProCTLA-4 cleavage occurs only within the TME requires additional validation. Is this metalloprotease induced in inflamed tissues?
4. The abstract claims that ProCTLA-4 is superior to currently used probodies in terms of toxicity, but this is not demonstrated in the manuscript.
5. One of the key findings of the manuscript is that ProCTLA-4 depletes Tregs in the tumor microenvironment. Data in Figures 3a and 3b show a 50% decrease in FoxP3+ cells within the CD4+ T cell population. However, it is unclear whether this effect is due to a decrease in FoxP3+ cells or an increase in FoxP3-CD4+ cells. This should be clarified.
6. Even if a 50% decrease in Tregs is observed, this reduction is much lower than that achieved by other antibodies designed to deplete Tregs. Thus, it is unlikely to have a significant impact on the antitumor effect. Moreover, the authors show that CD4+ T cell depletion before treatment does not impact the antitumor response (Figure 5a). How do the authors reconcile the lack of impact of CD4 depletion with the proposed mechanism of action on Tregs?
7. Can the Fc region of the antibody engage mouse Fc receptors? The sequence of the construct should be reported to enable other researchers to reproduce the data.

Minor Comments:

1. The manuscript's English should be revised for readability.
2. Introduction: The authors speculate about the withdrawal of BMS-986288 but provide no direct reference.
3. The abstract begins with a statement linking CTLA-4 expression on Tregs to its relevance in cancer therapy. However, the role of Tregs in the antitumor effect of clinically approved anti-CTLA-4 mAbs remains controversial. This sentence should be rewritten.
4. In the introduction, the authors state that 60% of patients undergoing anti-CTLA-4 mAb treatment experience grade 3-4 adverse effects. Please provide references for this statement.
5. The manuscript also claims that 90% of patients who respond to anti-CTLA-4 therapy achieve prolonged survival for over ten years. Please provide a reference for this statement.
6. Define the NSS mouse model.
7. In Figure 1d, the ProCTLA-4 outcome is not visible.
8. How did the authors identify CD4+ cells in the t-SNE analysis in Figure 3e?

9. Several panels across different figures do not follow a standard order from top to bottom and left to right.
10. Extended Figure 1d: Only one independent experiment should be shown.

Reviewer #2

(Remarks to the Author)

Nature Communications manuscript NCOMMS-11893

Reviewer comments

Cao et al. have constructed and tested a CTLA4 blocking antibody that has conditional binding upon activation by matrix metalloproteinase activity. This molecule, ProCTLA-4, upon cleavage, has much higher affinity for CTLA4 than the pro form and has comparable affinity for its target to ipilimumab (ipi). ProCTLA-4 accumulates in the tumors of syngeneic mice and has an apparent longer residence time in the tumor. ProCTLA-4 has good efficacy and is potentially more active than ipi in stimulating anti-cancer immune responses in vivo. ProCTLA-4, like other ipi-derivates, induces changes in T cell subsets within the tumor including the decrease in Treg cells and increases in CD4+ cells (presumably also increases in CD8+ cells or at least increased CD8+/Treg cell ratios). Their approach appears differentiated from other masking technologies, and they provided a direct comparison to the CytomX masked version of ipilimumab, BMS-986288.

Although the approach is novel, it is not clear if these results alone represent findings that deserve publication in Nature communications without further support and discussion. There are some major issues in the data presentation used to support the concept that the ProCTLA-4 antibody does not induce CRS leading to toxicities they observed with ipilimumab (see below). In addition, the conclusions that Pro-CTLA-4 has better efficacy ipi is not supported by pharmacological data showing a clear exposure to response relationship demonstrating increased efficacy. In essence, differences in the PK of ProCTLA-4 vs ipi in mice could account for the differences in efficacy and tumor residence time, which could be due to the construct mass and size. In the syngeneic tumor models, different flat doses were given depending on the models used (40, 100, 200 ug). For individual tumor models it would be good to see a dose response and calculate the EC50s for tumor killing to get a better idea of potency for ProCTLA-4 and ipi.

The following major points should be addressed in a revised manuscript before considering for publication:

1. What anti-CTLA4 antibody is being used to construct ProCTLA-4? I am assuming it is ipilimumab, but there are little details of its structure and epitope. Note that the epitope for ipi has been well studied, particularly regarding understanding efficacy. It is not entirely clear to the reader that ProCTLA-4 is indeed a modified derivative of ipilimumab. This should be clarified and more details about the construct and its CTLA-4 binding properties (i.e., epitope and affinity) should be provided. In a recent article (Robison et al, mAbs, 17:1, 2451296, DOI: 10.1080/19420862.2025.2451296) the ipi-epitope on human CTLA-4 has been extensively characterized and variants were generated that cross-react with the same epitope. They also show that the classical murine surrogate antibody 9D9, which binds mouse CTLA-4 and is a commonly used surrogate for CTLA-4 checkpoint blockade studies in mouse cancer models. In this work, it was shown that 9D9 has significant biophysical dissimilarities to therapeutic CTLA-4 antibodies and the 9D9-mCTLA4 complex crystal structure shows that the surrogate antibody binds an epitope distinct from ipilimumab and tremelimumab. In addition, while ipilimumab has pH-independent binding to hCTLA-4, 9D9 loses binding to mCTLA-4 at physiologically relevant acidic pH ranges. It is not clear which antibody was used to produce the mProCLTA-4, but the nature and epitope binding of it to CTLA-4 is an important consideration and could impact the immunophenotyping data that is provided.

2. A major thesis of this work is that the ProCLTA-4 antibody has less toxicity, and this will translate to an improved anti-CTLA-4 treatment either as a monotherapy or combination therapy. With respect to toxicities observed by ipi and not observed by ProCTLA4 in the mouse xenograft tumor model shown, it is only observed in immunodeficient (NNS) mice flooded with human PBMCs and not demonstrated in the syngeneic huCTLA4 KI mouse models. In the T-cell flooding experiments, the addition of human PBMC followed by treatments with ipi results in increased INFg levels as a surrogate marker of CRS. Does this occur syngeneic hu CTLA-4 KI mice as well? One explanation is that the addition of ipi to tumor bearing mice inoculated with human PBMCs accelerates graft versus host disease that eventually occurs in this model, and this is not a surrogate measure of the immunotoxicities observed in patients treated with ipi. In the clinic, ipi does not induce CRS, or it is rarely observed. The main toxicity that plagues this treatment is autoimmune tissue damage that occurs 2-3 months after treatment initiation, such as colitis and pancreatitis for examples. This toxicity is exacerbated if given in combination with PD-1 blockade. Although there are infusion related reactions that occur early after administration that are treated with diphenhydramine medication, this is clinically distinct from CRS and does not affect patient compliance. In the model the authors used, ProCTLA-4 does not, or minimally induces INFg levels as one would expect based on the masking structure employed. These data therefore may not translate to improved safety of ProCTLA-4 in the clinic.

Reviewer #3

(Remarks to the Author)

This study presents ProCTLA-4, a novel antibody version of anti-CTLA-4 therapy designed to mitigate toxicity while enhancing anti-tumor efficacy. Unlike prior approaches (e.g., BMS-986288), ProCTLA-4 employs a tumor-associated protease-cleavable Fab conformation instead of a masking peptide, reducing systemic immune-related adverse events (irAEs) while preserving efficacy in the tumor microenvironment (TME). It preferentially depletes ICOS-high Tregs, shifting the TME toward an anti-tumor state and enhancing antigen-specific CD8+ T cell responses. Additionally, ADCC-enhanced ProCTLA-4 demonstrates superior tumor control without increased toxicity. The study includes comprehensive preclinical testing across multiple tumor models (MC38, B16F10, humanized PBMC models), with scRNA-seq profiling, ELISpot

assays, and rechallenge experiments to assess memory response.

It is an interesting study. Some concerns:

- The study lacks a direct comparison between ProCTLA-4 and other engineered CTLA-4 therapies
- Toxicity and safety evaluations are limited to short-term assessments, with no data on long-term autoimmunity, anti-drug antibody (ADA) responses, or systemic exposure (PK).
- The mechanism behind ProCTLA-4's selective depletion of ICOS-high Tregs remains unclear, as no FcγR engagement assays were performed to confirm whether this effect is primarily ADCC-mediated.
- Certain data, such as flow cytometry gating strategies and Western blot images, lack full raw data availability.

Version 1:

Reviewer comments:

Reviewer #1

(Remarks to the Author)

The authors have addressed all previous comments and concerns raised in the earlier review.

Reviewer #2

(Remarks to the Author)

Upon secondary review of the manuscript "A next generation antibody for anti-CTLA-4 antibody effectively mitigates toxicity and facilitates the anti-tumor efficacy", I find that the authors have addressed my two major points of concern in their revision. I find the revised version acceptable for publication

Reviewer #3

(Remarks to the Author)

A very well done revision, no further comments

RESPONSE TO REVIEWERS' COMMENTS

Dear Reviewers:

We are sincerely grateful to the insightful and constructive comments and suggestions provided regarding our manuscript titled "A next generation probody for anti-CTLA-4 antibody effectively mitigates toxicity and facilitates the anti-tumor efficacy" (Manuscript ID: NCOMMS-25-11893).

We have addressed almost all the concerns raised and refined the language for greater clarity and precision. Here, we present a detailed point-by-point response to the referees' comments, with the corresponding revisions clearly highlighted in the revised manuscript for easy reference.

It is sure that these revisions have substantially improved the manuscript in more conclusiveness and significance.

Sincerely,

Yang-Xin Fu

2025.07.15

Response to Reviewers:

Reviewer #1 (Remarks to the Author):

In this manuscript, Weian Cao et al. present a novel probody design (ProCTLA-4) for anti-CTLA-4 therapy, aiming to enhance tumor-specific activation while reducing systemic toxicity. By linking the VH domain to the CL domain via an MMP-cleavable linker, the authors introduce an alternative strategy to traditional peptide-masked probody therapeutics (e.g., BMS-986288). The study demonstrates a reduced percentage of FoxP3+ cells within the CD4+ T cell population in the tumor microenvironment (TME), particularly targeting ICOS-high Tregs, and highlights the superior anti-tumor efficacy of ProCTLA-4 compared to Ipilimumab. Additionally, an ADCC-enhanced version (ProCTLA-4ADCC) is shown to further improve efficacy without increasing toxicity. While the antibody format is novel, certain aspects require further clarification and validation:

Major Comments:

1. The manuscript highlights the induction of memory-like CD8+ T cells, but it does not indicate when the mice were rechallenged in Figure 5g. This information is crucial for evaluating whether the experiment is assessing immune memory.

Thank you for the comments. MC38 tumor-bearing mice were i.p. treated with antibodies and the cured mice were rechallenged with 8 times dose of MC38 tumor cells at 8 weeks post the initial tumor inoculation. To determine whether ProCTLA4 treatment can induce a potent memory response, we challenged the mice with MC38 and treated them with CTLA-4 or ProCTLA-4. Eight weeks later, we found that the ProCTLA-4 group had more antigen-specific memory CD8+ T cells than the CTLA-4 group in the lymph node (Fig. 5h,i). It indicates that mProCTLA-4 treatment can induce a stronger memory response compared to mCTLA-4 treatment.

2. While ProCTLA-4 demonstrates slightly higher antitumor activity in an in vivo MC38 experiment,

its advantage over BMS-986288 remains unclear since both share the same mechanism of action. It is unlikely that this novel format can overcome the limitations that led to the discontinuation of BMS-986288.

We appreciate the comments. Concerning that MC38 tumor model strongly responds to anti-CTLA4 therapy, we have compared the anti-tumor efficacy between BMS-986288 and ProCTLA-4 the B16F10 tumor model which poorly responded to anti-CTLA-4 or anti-PD-1 therapy. ProCTLA-4 treatment showed potent tumor controlling, but the BMS-986288 failed (Fig. 6c). This indicates that ProCTLA-4 antibody can effectively overcome the ICB-resistant tumor than BMS-986288.

3. The assumption that ProCTLA-4 cleavage occurs only within the TME requires additional validation. Is this metalloprotease induced in inflamed tissues?

Thank you for the comments. Metalloprotease (MMP14) is highly expressed in tumor tissues (tumor cells and tumor associated macrophages) compared to normal tissues¹. MMP-cleavable linker was applied in many anti-tumor pro-drugs such as anti-CD3 antibody, IL12, IL2, and IL15²⁻⁵. Based on these, the affinity of ProCTLA-4 can be recovered within the TME while remaining safe in the normal tissues. Besides tumor tissues, metalloprotease is also reported of increased expression in inflamed tissue, such as arthritis⁶.

4. The abstract claims that ProCTLA-4 is superior to currently used probodies in terms of toxicity, but this is not demonstrated in the manuscript.

Thank you for the advice. We have modified the statement of ProCTLA-4 advantages in the abstract, to make it more accurate.

5. One of the key findings of the manuscript is that ProCTLA-4 depletes Tregs in the tumor microenvironment. Data in Figures 3a and 3b show a 50% decrease in FoxP3+ cells within the CD4+ T cell population. However, it is unclear whether this effect is due to a decrease in FoxP3+ cells or an increase in FoxP3-CD4+ cells. This should be clarified.

We are grateful to the suggestions. We have re-calculated the absolute number of Treg and cCD4⁺ T cells within tumor upon ProCTLA-4 treatment and found that mProCTLA-4 treatment decreased the absolute number of Treg especially ICOS^{hi} Treg compared with mCTLA-4, but have not increased the number of conventional CD4⁺ T cells (Foxp3- CD4⁺ T cells). (Fig. 3k, l, Extended Fig. 3b)

6. Even if a 50% decrease in Tregs is observed, this reduction is much lower than that achieved by other antibodies designed to deplete Tregs. Thus, it is unlikely to have a significant impact on the antitumor effect. Moreover, the authors show that CD4⁺ T cell depletion before treatment does not impact the antitumor response (Figure 5a). How do the authors reconcile the lack of impact of CD4 depletion with the proposed mechanism of action on Tregs?

Thank you for the comments. To exclude the impact of conventional CD4⁺ T cells (cCD4), we have transferred OT1 cells with Treg to Rag1^{-/-} mice upon ProCTLA-4 treatment. Without the cCD4⁺ T cells, ProCTLA-4 could also show sufficient tumor controlling, which indicated that the efficacy of ProCTLA-4 was depending on Treg existent (Fig. 3m). We assumed that the cCD4⁺ T cells could play some supportive role on CD8⁺ T cells response⁷, which resulted in offsetting effect in total CD4⁺ T cells depletion upon treatment.

7. Can the Fc region of the antibody engage mouse Fc receptors? The sequence of the construct should be reported to enable other researchers to reproduce the data.

We appreciate the suggestion. The mouse FcγRs was reported to recognize human Fc and mediate its downstream function⁸. In addition, we have showed the sequence and marked the motifs of human and mouse version of ProCTLA-4 in Extended Fig. 1.

Minor Comments:

1. The manuscript's English should be revised for readability.

Thank you for the advice. We have revised the manuscript in terms of language for clarity and grammatical accuracy.

2. Introduction: The authors speculate about the withdrawal of BMS-986288 but provide no direct reference.

Thank you for the comments. Our statement on “withdrawal of BMS-986288” made the misunderstanding. We will remove this sentence. The phase II clinical trial of BMS-986288 has completed in Oct. 2024 (<https://clinicaltrials.gov/show/NCT03994601>). However, it was not intended to continue the further development of BMS-986288 beyond the phase II study⁹.

3. The abstract begins with a statement linking CTLA-4 expression on Tregs to its relevance in cancer therapy. However, the role of Tregs in the antitumor effect of clinically approved anti-CTLA-4 mAbs remains controversial. This sentence should be rewritten.

We appreciate the suggestion. We have modified the statement at the beginning of abstract to avoid unnecessary controversy.

4. In the introduction, the authors state that 60% of patients undergoing anti-CTLA-4 mAb treatment experience grade 3-4 adverse effects. Please provide references for this statement.

5. The manuscript also claims that 90% of patients who respond to anti-CTLA-4 therapy achieve

prolonged survival for over ten years. Please provide a reference for this statement.

Thank you for the advices. We have supplemented the references accurately at the statement.

6. Define the NSS mouse model.

Many thanks for the suggestion. We have supplemented the source of NSS mice in the Methods. NSG-SGM3 (NSS) mice were purchased from the Jackson Laboratory (#013062).

7. In Figure 1d, the ProCTLA-4 outcome is not visible.

Thank you for the comment. We have modified the figure legend in Fig. 1d to make it more readable.

8. How did the authors identify CD4+ cells in the t-SNE analysis in Figure 3e?

We appreciate the comments. We have identified Treg cells from the T/NK cell population by their co-expression of CD3e, CD4, and Foxp3 in the single-cell sequencing analysis. We have clarified the strategy in the figure legend.

9. Several panels across different figures do not follow a standard order from top to bottom and left to right.

Thank you for the advice. We have modified the layout of some figures to make it more readable.

10. Extended Figure 1d: Only one independent experiment should be shown.

Many thanks for the comment. We have modified the Extended Fig.1 to keep one independent experiment.

Reviewer #2 (Remarks to the Author):

Nature Communications manuscript NCOMMS-11893

Reviewer comments

Cao et al. have constructed and tested a CTLA4 blocking antibody that has conditional binding upon activation by matrix metalloproteinase activity. This molecule, ProCTLA-4, upon cleavage, has much higher affinity for CTLA4 than the pro form and has comparable affinity for its target to ipilimumab (ipi). ProCTLA-4 accumulates in the tumors of syngeneic mice and has an apparent longer residence time in the tumor. ProCTLA-4 has good efficacy and is potentially more active than ipi in stimulating anti-cancer immune responses in vivo. ProCTLA-4, like other ipi-derivates, induces changes in T cell subsets within the tumor including the decrease in Treg cells and increases in CD4+ cells (presumably also increases in CD8+ cells or at least increased CD8+/Treg cell ratios). Their approach appears differentiated from other masking technologies, and they provided a direct comparison to the CytomX masked version of ipilimumab, BMS-986288.

Although the approach is novel, it is not clear if these results alone represent findings that deserve publication in Nature communications without further support and discussion. There are some major issues in the data presentation used to support the concept that the ProCTLA-4 antibody does not induce CRS leading to toxicities they observed with ipilimumab (see below). In addition, the conclusions that Pro-CTLA-4 has better efficacy ipi is not supported by pharmacological data showing a clear exposure to response relationship demonstrating increased efficacy. In essence, differences in the PK of ProCTLA-4 vs ipi in mice could account for the differences in efficacy and tumor residence time, which could be due to the construct mass and size. In the syngeneic tumor models, different flat doses were given depending on the models used (40, 100, 200 ug). For individual tumor models it would be good to see a dose response and calculate the EC50s for tumor killing to get a better idea of potency for ProCTLA-4 and ipi.

We are very grateful to the suggestions. To compare the pharmacokinetics of ProCTLA-4 and Ipilimumab, ProCTLA-4 and Ipilimumab were administered intraperitoneally to hCTLA4 KI mice and the concentration of antibodies in serum were monitored along within 12 days. As shown in Extended Fig.1g, ProCTLA-4 arrived to the peak concentration quickly in 3 hours and retained at the peak concentration for much longer time than Ipi did in 6 days.

According to the suggestion, we have titrated the doses of ProCTLA-4 in MC38 tumor model and calculated the EC50 of ProCTLA-4 in tumor killing in vivo in about 40 µg (Extended Fig.

2). Unfortunately, the hCTLA-4 KI mice are not easy to generate well, so that we have not enough mice to set up more groups with Ipi treatment to calculate the EC50. However, according to our previous experiments with Ipi treatment in 100 μg (Fig.2a, 4b, 6b), we found that just 17.4% (4/23) of the mice could relapse from Ipi treatment, indicating the EC50 of Ipi might be over 100 μg .

The following major points should be addressed in a revised manuscript before considering for publication:

1. What anti-CTLA4 antibody is being used to construct ProCTLA-4? I am assuming it is ipilimumab, but there are little details of its structure and epitope. Note that the epitope for ipi has been well studied, particularly regarding understanding efficacy. It is not entirely clear to the reader that ProCTLA-4 is indeed a modified derivative of ipilimumab. This should be clarified and more details about the construct and its CTLA-4 binding properties (i.e., epitope and affinity) should be provided. In a recent article (Robison et al, mAbs, 17:1, 2451296, DOI: 10.1080/19420862.2025.2451296) the ipi-epitope on human CTLA-4 has been extensively characterized and variants were generated that cross-react with the same epitope. They also show that the classical murine surrogate antibody 9D9, which binds mouse CTLA-4 and is a commonly used surrogate for CTLA-4 checkpoint blockade studies in mouse cancer models. In this work, it was shown that 9D9 has significant biophysical dissimilarities to therapeutic CTLA-4 antibodies and the 9D9-mCTLA4 complex crystal structure shows that the surrogate antibody binds an epitope distinct from ipilimumab and tremelimumab. In addition, while ipilimumab has pH-independent binding to hCTLA-4, 9D9 loses binding to mCTLA-4 at physiologically relevant acidic pH ranges. It is not clear which antibody was used to produce the mProCLTA-4, but the nature and epitope binding of it to CTLA-4 is an important consideration and could impact the immunophenotyping data that is provided.

Thank you for the comments. ProCTLA-4 is designed based on the sequence of Ipi, which we have clarified in the introduction. In addition, we have provided the sequence and marked the motifs of human and mouse version of ProCTLA-4 in Extended Fig. 1. As for the mProCTLA-4, we have used clone 4F10 to construct the mouse version, which was reported as the CTLA-4 depleting antibody¹⁰. Unlike 9D9, there is no research paper decoding the crystal structure and epitope of this clone.

a

Sequence of ProCTLA-4 (human version)

b

Sequence of mProCTLA-4 (mouse version)

2. A major thesis of this work is that the ProCTLA-4 antibody has less toxicity, and this will translate to an improved anti-CTLA-4 treatment either as a monotherapy or combination therapy. With respect to toxicities observed by ipi and not observed by ProCTLA4 in the mouse xenograft tumor model shown, it is only observed in immunodeficient (NBS) mice flooded with human PBMCs and not demonstrated in the syngeneic huCTLA4 KI mouse models. In the T-cell flooding experiments, the addition of human PBMC followed by treatments with ipi results in increased INF γ levels as a surrogate marker of CRS. Does this occur syngeneic hu CTLA-4 KI mice as well? One explanation is that the addition of ipi to tumor bearing mice inoculated with human PBMCs accelerates graft versus host disease that eventually occurs in this model, and this is not a surrogate measure of the immunotoxicities observed in patients treated with ipi. In the clinic, ipi does not induce CRS, or it is rarely observed. The main toxicity that plagues this treatment is autoimmune tissue damage that occurs 2-3 months after treatment initiation, such as colitis and pancreatitis for examples. This toxicity is exacerbated if given in combination with PD-1 blockade. Although there are infusion related reactions that occur early after administration that are treated with diphenhydramine medication, this is clinically distinct from CRS and does not affect patient compliance. In the model the authors used, ProCTLA-4 does not, or minimally induces INF γ levels as one would expect based on the masking structure employed. These data therefore may not translate to improved safety of ProCTLA-4 in the clinic.

Thank you for the insightful suggestion. We agree that it is very important to evaluate the toxicity of ProCTLA-4 in various types of models. To test the potential autoimmune tissue damage, we compared ProCTLA-4 and Ipilimumab in young hCTLA4-KI mice combined with anti-PD-1 treatment to sensitize the mice to irAE as reported before¹¹. With anti-PD-1 combination, Ipi induced severe body weight loss and high level of TNF α in serum, but ProCTLA-4 did not. Furthermore, Ipi elicited severe multi-tissue inflammation, which was manifested as pancreatitis, colitis and pulmonary fibrosis and hemorrhage. In contrast, ProCTLA-4 exhibited better safety and improved tissue inflammation (Fig. 1f-i).

Reviewer #3 (Remarks to the Author):

This study presents ProCTLA-4, a novel probody version of anti-CTLA-4 therapy designed to mitigate toxicity while enhancing anti-tumor efficacy. Unlike prior approaches (e.g., BMS-986288), ProCTLA-4 employs a tumor-associated protease-cleavable Fab conformation in-stead of a masking peptide, reducing systemic immune-related adverse events (irAEs) while preserving efficacy in the tumor microenvironment (TME). It preferentially depletes ICOS-high Tregs, shifting the TME toward an anti-tumor state and enhancing antigen-specific CD8+ T cell responses. Additionally, ADCC-enhanced ProCTLA-4 demonstrates superior tumor control without increased toxicity. The study includes comprehensive preclinical testing across multiple tumor models (MC38, B16F10, humanized PBMC models), with scRNA-seq profiling, ELISpot assays, and rechallenge experiments to assess memory response.

It is an interesting study. Some concerns:

- The study lacks a direct comparison between ProCTLA-4 and other engineered CTLA-4 therapies

We appreciate the comments and further communications with editor and you to clarify the issue. We have indeed compared the efficacy and toxicity with Ipi and also well-known engineered CTLA-4 (BMS-986288) to our ProCTLA-4. Ipi and BMS-986288 are best known anti-CTLA4 and its pro-drug. In addition, we have evaluated the advantages of ProCTLA4 over BMS-986288 in the B16F10 tumor model, which showed better efficacy of ProCTLA-4 in the ICB-resistant model (Fig. 6c).

- 2. Toxicity and safety evaluations are limited to short-term assessments, with no data on long-term autoimmunity, anti-drug antibody (ADA) responses, or systemic exposure (PK).

Thank you for the important suggestions. We agree that it is very important to evaluate the toxicity of ProCTLA-4 in a long-term model. To test the potential autoimmune tissue damage, we compared ProCTLA-4 and Ipilimumab in young C57 background hCTLA4-KI mice combined with anti-PD-1 treatment to sensitize the mice to irAE as reported before¹¹. With anti-PD-1 combination, Ipi induced severe body weight loss and high level of TNF α in serum, but ProCTLA-4 did not. Furthermore, Ipi elicited severe multi-tissue inflammation, which was manifested as pancreatitis, colitis and pulmonary fibrosis and hemorrhage. In contrast, ProCTLA-4 exhibited better safety and improved tissue inflammation (Fig. 1f-i).

To compare the pharmacokinetics of ProCTLA-4 and Ipilimumab, ProCTLA-4 and Ipilimumab were administered intraperitoneally to hCTLA4 KI mice and the concentration of antibodies in serum were monitored along within 12 days. As shown in Extended Fig.1g, ProCTLA-4 arrived at the peak quickly in several hours but retained at the peak concentration for much longer time than Ipi did in 6 days.

- 3. The mechanism behind ProCTLA-4's selective depletion of ICOS-high Tregs remains unclear, as no Fc γ R engagement assays were performed to confirm whether this effect is primarily ADCC-mediated.

We are grateful to the advice. To determine whether the depletion of ICOS-high Tregs depends on FcγR engagement, we constructed mProCTLA-4 with a mutant Fc that cannot be recognized by FcR (mProCTLA-4_{no ADCC})¹². ProCTLA-4_{no ADCC} failed to control MC38 tumor growth and showed no effect on ICOS-high Treg depletion. These data indicate that ICOS-high Treg depletion depends on FcγR engagement (Extended Fig. 3c-d).

- Certain data, such as flow cytometry gating strategies and Western blot images, lack full raw data availability.

Thank you for the comments. We have provided the flow gating strategies in Extended Fig.3a. All the raw data including the gel images are shown in a separated file.

(1) Gating strategy:

(2) SDS-PAGE raw data of Extended Fig. 1d:

Lane M: molecular mass makers
 Lane 1: Reducing ProCTLA-4
 Lane 2: Non-reducing ProCTLA-4
 Lane 3: Reducing ProCTLA-4(MMP14 Cleaved)
 Lane 4: Reducing ProCTLA-4(without MMP14)

Reference:

- 1 Kessenbrock, K., Plaks, V. & Werb, Z. Matrix metalloproteinases: regulators of the tumor microenvironment. *Cell* **141**, 52-67, doi:10.1016/j.cell.2010.03.015 (2010).
- 2 Trang, V. H. *et al.* A coiled-coil masking domain for selective activation of therapeutic antibodies. *Nature biotechnology* **37**, 761-765, doi:10.1038/s41587-019-0135-x (2019).
- 3 Mansurov, A. *et al.* Masking the immunotoxicity of interleukin-12 by fusing it with a domain of its receptor via a tumour-protease-cleavable linker. *Nat Biomed Eng* **6**, 819-829, doi:10.1038/s41551-022-00888-0 (2022).
- 4 Zhang, X. *et al.* TIM3-blockade synergizes with IL2 in alleviating intra-tumoral CD8(+)T cell exhaustion. *Nat Commun* **16**, 5130, doi:10.1038/s41467-025-60463-4 (2025).
- 5 Guo, J. *et al.* Tumor-conditional IL-15 pro-cytokine reactivates anti-tumor immunity with limited toxicity. *Cell research* **31**, 1190-1198, doi:10.1038/s41422-021-00543-4 (2021).
- 6 Parks, W. C., Wilson, C. L. & López-Boado, Y. S. Matrix metalloproteinases as modulators of inflammation and innate immunity. *Nat Rev Immunol* **4**, 617-629, doi:10.1038/nri1418 (2004).
- 7 van der Heide, V. *et al.* Prolonged but finite antigen presentation promotes reversible defects of "helpless" memory CD8(+) T cells. *Immunity* **58**, 1742-1761.e1714, doi:10.1016/j.immuni.2025.05.025 (2025).
- 8 Dekkers, G. *et al.* Affinity of human IgG subclasses to mouse Fc gamma receptors. *mAbs* **9**, 767-773, doi:10.1080/19420862.2017.1323159 (2017).
- 9 Therapeutics, C. *2024 Annual Report*, <<https://ir.cytomx.com/static-files/1494856d-d28f-4de4-9cb8-be42c12a1a7c>> (2024).
- 10 Loughry, A., Fairchild, S., Athanasou, N., Edwards, J. & Hall, F. C. Inflammatory arthritis and dermatitis in thymectomized, CD25+ cell-depleted adult mice. *Rheumatology (Oxford, England)* **44**, 299-308, doi:10.1093/rheumatology/keh477 (2005).
- 11 Zhang, Y. *et al.* Hijacking antibody-induced CTLA-4 lysosomal degradation for safer and more effective cancer immunotherapy. *Cell research* **29**, 609-627, doi:10.1038/s41422-019-0184-1 (2019).
- 12 Schlothauer, T. *et al.* Novel human IgG1 and IgG4 Fc-engineered antibodies with completely abolished immune effector functions. *Protein engineering, design & selection : PEDS* **29**, 457-

466, doi:10.1093/protein/gzw040 (2016).

RESPONSE TO REVIEWERS' COMMENTS

Reviewer #1 (Remarks to the Author):

The authors have addressed all previous comments and concerns raised in the earlier review.

Thank you very much for your constructive comments, which has greatly contributed to improving the quality of our work.

Reviewer #2 (Remarks to the Author):

Upon secondary review of the manuscript "A next generation probody for anti-CTLA-4 antibody effectively mitigates toxicity and facilitates the anti-tumor efficacy", I find that the authors have addressed my two major points of concern in their revision. I find the revised version acceptable for publication

We greatly appreciate your recognition of the revisions we made to address your concerns, and we are grateful for your recommendation for publication.

Reviewer #3 (Remarks to the Author):

A very well done revision, no further comments.

Thank you very much for your positive assessment of our revised manuscript.